# Studies of the horizontal inhomogeneities in $NO_2$ concentrations above a shipping lane using ground-based MAX-DOAS measurements and validation with airborne imaging DOAS measurements

André Seyler[1], Andreas C. Meier[1], Folkard Wittrock[1], Lisa Kattner[1,2], Barbara Mathieu-Üffing[1,2,a], Enno Peters[1,b], Andreas Richter[1], Thomas Ruhtz[3], Anja Schönhardt[1], Stefan Schmolke[2], and John P. Burrows[1]

[1]Institute of Environmental Physics, University of Bremen, Germany
[2]Federal Maritime and Hydrographic Agency (BSH), Hamburg, Germany
[3]Institute for Space Sciences, Freie Universität Berlin, Germany
[a]now at: State Agency for Agriculture, Environment and Rural Areas Schleswig-Holstein (LLUR), Germany
[b]now at: Institute for the Protection of Maritime Infrastructures, German Aerospace Center (DLR), Bremerhaven, Germany

*Correspondence to:* André Seyler (aseyler@iup.physik.uni-bremen.de)

**Abstract.**

This study describes a novel application of an "onion peeling" like approach to MAX-DOAS measurements of shipping emissions aiming at investigating the strong horizontal inhomogeneities in $NO_2$ over a shipping lane. To monitor ship emissions on the main shipping route towards the port of Hamburg, a two-channel (UV and visible) MAX-DOAS instrument was deployed on the island Neuwerk in the German Bight, 6–7 km south of the main shipping lane. Utilizing the fact that the effective light path length in the atmosphere depends systematically on wavelength, simultaneous measurements and DOAS retrievals in the UV and visible spectral range are used to probe air masses at different horizontal distances to the instrument to estimate two-dimensional pollutant distributions. Two case-studies have been selected to demonstrate the ability to derive the approximate plume positions in the observed area. A situation with northerly wind shows high $NO_2$ concentrations close to the measurement site and low values in the north of the shipping lane. The opposite situation with southerly wind, unfavorable for the on-site in situ instrumentation, demonstrates the ability to detect enhanced $NO_2$ concentrations several kilometers away from the instrument. Using a Gaussian plume model, in-plume $NO_2$ volume mixing ratios can be derived from the MAX-DOAS measurements.

For validation, a comparison to airborne imaging DOAS measurements during the NOSE campaign in July 2013 is performed, showing good agreement between the approximate plume position derived from the onion peeling MAX-DOAS and the airborne measurements as well as between the derived in-plume $NO_2$ VMRs.

# 1 Introduction

Over the last decades, there has been a strong increase in ship traffic and shipping emissions of gas phase pollutants but a reduction in their land sources in much of Europe. This has lead to an increasing contribution of shipping emissions to air pollution in coastal regions. Consequently, emission reduction measures have been enacted by the International Maritime Organization (IMO) in the International Convention for the Prevention of Pollution from Ships (MARPOL 73/78 Annex VI) globally as well as, more stringent, locally in so-called emission control areas (ECAs) like North and Baltic Sea (IMO, 2009). To reduce sulfur oxides ($SO_x$) emissions, at the time of this study, the allowed sulfur content in shipping fuel is limited to 0.1 % in ECAs (since 2015, before: 1.0 %) and to 3.5 % globally, which is planned to be reduced to 0.5 % by 2020. For $NO_x$, the allowed emission rate depends on the rated rotational speed of the engine crankshaft (engine power and fuel efficiency) and is implemented in 3 tiers: Tier I (globally) for ships built between 2000 and 2010, Tier II (globally) for ships built from 2011 onwards and Tier III (locally in ECAs) for ships built from 2016 onwards, the last one not yet implemented in North and Baltic sea, shifted to 2021 (IMO, 2017). In order to monitor the effectiveness of these measures as well as the overall impact of ship emissions on air quality, measurements of air pollution from ships are required.

Most measurements of air pollution are performed with in situ instrumentation, and this includes monitoring of the effect of ship emissions, which is usually performed with either land-based or shipborne in situ measurements. As shown in Seyler et al. (2017), MAX-DOAS measurements can provide both a complementary approach and an alternative to in situ trace gas measurements at sites, where the ships are several kilometers away from the instrument and interpretation of in situ measurements is challenging due to dilution and broadening of the plume during the travel time from the ships to the measurement site.

MAX-DOAS measurements pointing at the horizon probe a long horizontal light path and are thus very sensitive to absorbers located close to the ground. The strong wavelength dependence of Rayleigh scattering ($\propto \lambda^{-4}$) leads to longer effective horizontal light paths for longer wavelengths. Simultaneous measurements and DOAS retrievals in the UV and visible spectral range can thus be used to probe different parts of the horizontal light path, an approach which is often called "onion peeling" method and has been applied to MAX-DOAS measurements before; Ortega et al. (2015) used this method to retrieve two dimensional $NO_2$ fields from circular azimuth scans around the instrument in the framework of the MAD-CAT campaign (Multi-Axis DOAS Comparison campaign for Aerosols and Trace gases) in Mainz, Germany. The aim of the study was the investigation of horizontal gradients in a strongly polluted urban area, with the cities of Mainz, Wiesbaden and Frankfurt as well as the Frankfurt airport close by, focussing on comparison to satellite measurements.

The present study focuses on measurements in a relatively clean coastal region where ships passing by the island are often the only dominant source of air pollution (Seyler et al., 2017). The ships are mobile point sources of $NO_x$ emissions and the emitted exhaust gas plumes are transported, depending on wind conditions, leading to a strongly inhomogeneous $NO_2$ field over the shipping lane.

Ortega et al. (2015) probed a circular area with 14 azimuthal viewing directions distributed over a 360° view around the instrument. In the present study, a similar measurement pattern was applied using 5 different azimuth directions distributed

over a 120° angle to cover the shipping lane close to the island (see Fig. 1b) with sufficient time resolution to monitor individual passing ships. The onion peeling approach provides additional distance information for the measured $NO_2$ columns.

This study uses measurements in both the UV ($\sim$350 nm) and blue spectral range ($\sim$450 nm), while Ortega et al. (2015) used additional measurements in the yellow spectral range ($\sim$570 nm) to get an even longer effective horizontal light path and cover a larger region. This is not possible here as the instrument used has a smaller wavelength coverage.

As can be seen from Fig. 1a and b, the measurement site on the island Neuwerk is ideal for applying this measurement principle: The distance between site and shipping lane is of the order of 6 to 10 kilometers, depending on the azimuthal viewing direction, which is in the range of typical UV horizontal effective light path lengths (Seyler et al., 2017). Depending on the azimuthal direction, the additional probing distance gained by measurements in the visible spectral range covers the shipping lane or the region in the north of the ship track. As it is shown in the following, this enables the $NO_2$ distribution caused by the ship emission plumes over and around the ship track to be determined. In addition even the distance and course of the emitted plumes is observed.

This publication is a follow up to an earlier study entitled "Monitoring shipping emissions in the German Bight using MAX-DOAS measurements" (Seyler et al., 2017) where long-term measurements were used to asses the impact of shipping emissions on the regional air quality, while the present study focuses on describing, demonstrating and validating a new method for improved measurements of ship emissions and their localization.

The present study is part of the project MESMART (measurements of shipping emissions in the marine troposphere), a cooperation between the University of Bremen (Institute of Environmental Physics, IUP) and the German Federal Maritime and Hydrographic Agency (Bundesamt für Seeschifffahrt und Hydrographie, BSH), supported by the Helmholtz Zentrum Geesthacht. For further information visit http://www.mesmart.de/.

## 2 Measurement site and instrumentation

### 2.1 MAX-DOAS instrument

The multi axis differential optical absorption spectroscopy (MAX-DOAS) (Hönninger et al., 2004; Wittrock et al., 2004) is a well-established technique for measurements of trace gases that absorb in the UV and visible spectral range. This passive remote sensing method measures spectra of scattered sunlight in multiple viewing directions and is highly sensitive to absorbers in the atmospheric boundary layer. A two-channel MAX-DOAS instrument was deployed on the island Neuwerk from July 2013 to July 2016. It comprises a telescope unit with a field of view of 1° on a pan-tilt head, an optical fiber cable and two spectrometers with CCD cameras for UV (304.6–371.7 nm) and visible (398.8–536.7 nm) spectral range. This arrangement is optimized for the simultaneous retrieval of $NO_2$ and $O_4$ in both spectral domains. The total exposure time (or integration time) per measurement is 10 seconds for off-axis measurements and 20 seconds for zenith sky reference measurements. A new azimuthal measurement in one of the five different directions (see Section 2.2 and Fig. 1) starts about every 30 seconds. The measurement sequence is intermitted by a vertical scan in the main direction (335° azimuth) and a zenith sky measurement,

**Table 1.** DOAS fit settings for the retrieval of $NO_2$ and $O_4$ in UV and visible spectral range

| Parameter | NO$_2$ (UV) | NO$_2$ (visible) |
|---|---|---|
| **Fitting window** | 338–370 nm | 425–497 nm |
| **Polynomial degree** | 4 | 3 |
| **Intensity offset** | Constant | Constant |
| **Zenith reference** | Coinciding zenith measurement* | Coinciding zenith measurement* |
| **SZA limit** | Up to 85° SZA | Up to 85° SZA |
| **O3** | 223 K & 243 K (Serdyuchenko et al., 2014) | 223 K (Serdyuchenko et al., 2014) |
| **NO$_2$** | 298 K (Vandaele et al., 1996) | 298 K (Vandaele et al., 1996) |
| **O$_4$** | 293 K (Thalman and Volkamer, 2013) | 293 K (Thalman and Volkamer, 2013) |
| **H2O** | – | 293 K (Lampel et al., 2015) |
| **HCHO** | 297 K (Meller and Moortgat, 2000) | – |
| **Ring** | SCIATRAN (Rozanov et al., 2014) | SCIATRAN (Rozanov et al., 2014) |

\* Interpolation in time between the zenith measurements directly before and after the off-axis scan.

both together taking in total around 90 seconds. The temporal resolution for one viewing direction, i.e. the time until the same azimuthal direction is probed again, is around 4 minutes.

A detailed description of the MAX-DOAS instrument and its components as well as the general measurement geometry for ship emission measurements is given in Seyler et al. (2017). Details of the DOAS fit settings used are summarized in Table 1.

## 2.2 Measurement site

Neuwerk is a small island in the German Bight, northwest of the city of Cuxhaven at the mouth of the river Elbe, around 9 kilometers off the coast. An overview of the area is shown in Fig. 1a. The main shipping lane into the river Elbe towards the port of Hamburg passes the island in the north at a distance of 6–7 km (see Fig. 1a). The MAX-DOAS instrument was installed on a radar tower at a height of 30 meters above ground level. Additional instrumentation on site included in situ gas analyzers ($NO_x$, $SO_2$, $O_3$, $CO_2$) in a combined compact housing (Airpointer from MLU-recordum, Austria), a Davis Vantage Pro 2 semi-professional weather station and an automatic identification system (AIS, (IMO, 2002)) receiver. The AIS signal broadcasts various information like identification, position, speed, course and size of the ship. Broadcasting equipment is mandatory for all ships larger than 20 m. In the present study, the AIS information is used to attribute the measurements to individual ships. Wind direction and speed is available with a time resolution of 10 minutes from two stations (see Fig. 1a), one on Neuwerk and one on the neighboring island Scharhörn, operated by the Hamburg Port Authority (HPA).

To sample a larger region, the MAX-DOAS was set up to have five different azimuthal viewing directions: 310°, 335°, 5°, 35° and 65° with respect to north, each pointing towards different sections of the shipping lane (see Fig. 1b).

For further information on the measurement site and instrumentation see Seyler et al. (2017).

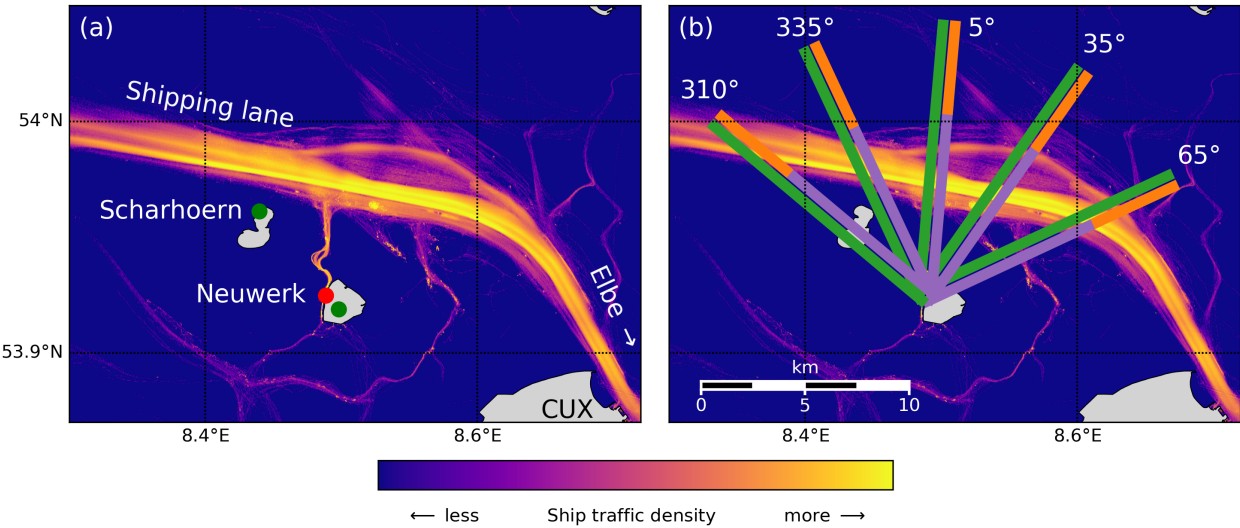

**Figure 1.** (a) Ship traffic density map calculated from all received AIS messages (2013-2016) showing the main shipping lane from the North sea into the Elbe river close to the measurement site on a radar tower on the island Neuwerk (red dot). Wind measurements are available on Neuwerk as well as the neighboring island Scharhörn (green dots). (b) Effective horizontal light paths in UV (purple line) and visible spectral range (green line) for the five azimuthal viewing directions of the MAX-DOAS instrument (310°, 335°, 5°, 35°, 65°, with respect to north), shown for typical light path lengths of $9\,\mathrm{km}$ (UV) and $13\,\mathrm{km}$ (vis), respectively. The difference between both paths, $\Delta L$, is highlighted by the orange line.

## 3   Methodology

The quantity retrieved from DOAS measurements is the so-called slant column density (SCD), the integrated concentration of an absorber along the atmospheric light path. To measure the $NO_2$ absorption inside the ship plumes emitted on the shipping lane, the instrument is pointing in 0.5° elevation towards the horizon. Taking a close-in-time zenith-sky measurement as a reference, in a first assumption only the absorption along the horizontal part of the effective light path is retrieved and the absorption higher up in the atmosphere cancels out. This yields the differential slant column density (DSCD).

For the comparison with in situ measurements the MAX-DOAS horizontal trace gas columns are converted to horizontal path averaged volume mixing ratios (VMR) by using the $O_4$ scaling approach (see Section 3.1). The onion peeling approach (see Section 3.2) is used to separate $NO_2$ absorptions at different horizontal distances to derive separate $NO_2$ VMRs and estimate the distance to the plumes.

### 3.1 $O_4$ scaling approach – methodology and limitations

The oxygen collision complex $O_4$ absorbs in similar wavelength ranges as $NO_2$ in the UV and visible. Since the near-surface concentration of $O_4$ is known, the effective horizontal path length can be calculated by dividing the DSCD of $O_4$ by its number density $n_{O_4}$:

$$5 \quad L = \frac{\text{SCD}_{O_4,\text{horiz}} - \text{SCD}_{O_4,\text{zenith}}}{n_{O_4}} = \frac{\text{DSCD}_{O_4}}{n_{O_4}} \tag{1}$$

with $n_{O_4} = (n_{O_2})^2$, which can be calculated from the measured temperature and pressure. This can be done independently for both UV and visible measurements, giving average light path lengths of $L_{\text{UV}} = (9.3 \pm 2.3)\,\text{km}$ and $L_{\text{vis}} = (12.9 \pm 4.5)\,\text{km}$ [mean $\pm$ standard deviation] for the three years of measurements on Neuwerk, depending on the observational conditions. Under clear sky conditions, typical light path lengths are $10\,\text{km}$ in the UV and $15\,\text{km}$ in the visible spectral range (Seyler et al.,
10   2017).

Knowing the horizontal light path length $L$, the $NO_2$ DSCD can be divided by $L$ to obtain the average concentration (number density) of $NO_2$ along the horizontal light path. Dividing the $NO_2$ concentration by the concentration of air, $n_{\text{air}}$, which can be calculated via the ideal gas law from the measured temperature and pressure, yields the average volume mixing ratio (VMR) along $L$:

$$15 \quad \text{VMR}_{NO_2} = \frac{\text{SCD}_{NO_2,\text{horiz}} - \text{SCD}_{NO_2,\text{zenith}}}{L \cdot n_{\text{air}}} = \frac{\text{DSCD}_{NO_2}}{L \cdot n_{\text{air}}} \tag{2}$$

This $O_4$ scaling approach has been successfully applied to MAX-DOAS measurements before, for example in urban polluted areas (Sinreich et al., 2013; Wang et al., 2014) or at high mountain sites (Gomez et al., 2014; Schreier et al., 2016).

For a homogeneous, well-mixed $NO_2$ field along the light path, this VMR must agree with in situ measurement from the same altitude. For the ship emission case, where emission plumes are filling only a small fraction of the several kilometers long
light path, the path-averaged MAX-DOAS VMR will not represent the VMR inside the plume and values will be smaller than in situ measurements inside the plume (Seyler et al., 2017).

In addition, the different shapes of the atmospheric profiles of $NO_2$ (emitted and formed close to the surface) and $O_4$ (exponentially decreasing with altitude) introduce systematic errors as has been shown by Sinreich et al. (2013) and Wang et al. (2014). To account for this, correction factors calculated by radiative transfer simulations are needed. These depend
on well-known quantities such as solar zenith angle (SZA) and relative solar azimuth angle (RSAA) as well as on unknown quantities such as aerosol optical density (AOD), height of the $NO_2$ box profile and the extent and vertical position of the aerosol layer relative to the $NO_2$ profile (Sinreich et al., 2013), which are not measured and cannot be easily approximated for the present study. In previous studies, it has been assumed that $NO_2$ is well mixed within a layer from the surface up to a top layer height and absent above this altitude. This is not a valid assumption in case of horizontally inhomogeneous $NO_2$
fields such as those probed over the shipping lane. As in Seyler et al. (2017), scaling factors are therefore not considered here,

presumably leading to a systematic overestimation of path lengths and thus underestimation of MAX-DOAS VMRs (Sinreich et al., 2013; Wang et al., 2014).

Clouds can decrease or increase the light path length (and $O_4$ absorption) by multiple scattering, depending on the cloud's position and its optical properties, especially its optical thickness (Wagner et al., 2014). As a result, a day with scattered or broken clouds will show much more variation in path lengths than a clear sky day, even between consecutive measurements, by having clouds in either off-axis or reference measurement or both or neither, which makes interpretation of results more difficult. In the following, only clear sky days or measurements under cloud free conditions are considered.

## 3.2 "Onion peeling" MAX-DOAS approach

As mentioned above, the wavelength dependence of Rayleigh scattering results in a wavelength dependence of the light path lengths after the last scattering point. This can be utilized to probe different air masses in the atmosphere by measuring both in the UV and visible spectral range.

The aforementioned $O_4$ scaling method gives two path-averaged volume mixing ratios for each measurement; one for the shorter UV and one for the longer visible effective horizontal light path, which are shown in Fig. 1b and 2 as a purple and green line, respectively. One can calculate a third volume mixing ratio from the difference of the two DSCDs and path lengths:

$$\text{VMR}_{@\Delta L} = \frac{\text{DSCD}_{\text{vis}} - \text{DSCD}_{\text{UV}}}{(L_{\text{vis}} - L_{\text{UV}}) \cdot n_{\text{air}}} = \frac{\Delta\text{DSCD}}{\Delta L \cdot n_{\text{air}}} \tag{3}$$

This yields the average volume mixing ratio $\text{VMR}_{@\Delta L}$ along the path difference $\Delta L$, which is shown as an orange line in Fig. 1b and 2.

As each ship is a moving point source for $NO_2$ emissions, the $NO_2$ field over a shipping lane is strongly inhomogeneous. This means that the $NO_2$ is in general not distributed evenly along any of the effective horizontal light paths.

Depending on the position of the plume in relation to the UV and visible light path, the path averaged mixing ratios can differ substantially. Figure 2 shows schematically the plume-light-path geometry for three possible observation scenarios and illustrates the expected $NO_2$ signal for the different horizontal light paths.

In case (a) the plume is close to the instrument and is completely covered by the shorter UV path $L_{\text{UV}}$, i.e. it is closer to the instrument than the (mean) last scattering point in the UV. Although both paths cover the same amount of $NO_2$, the retrieved path-averaged concentration is higher for the UV signal because of the higher relative contribution of the fraction of the light path which probes the $NO_2$ plume. The path difference $\Delta L$ incorporates no $NO_2$ from the emission plume, resulting in zero or background level $NO_2$ from there. It can be seen from Fig. 1b that this situation occurs for northerly wind directions. Section 4.2 shows example measurement results for such a case.

Case (b) shows the opposite situation, when the plume is further away from the instrument than the UV scattering point and only covered by the visible path $L_{\text{vis}}$. This results in an enhanced signal for the $NO_2$ retrieved in the visible, and no signal in the UV. The path averaged concentration retrieved for $\Delta L$ is even higher, because $\Delta L$ is only a segment of the visible path

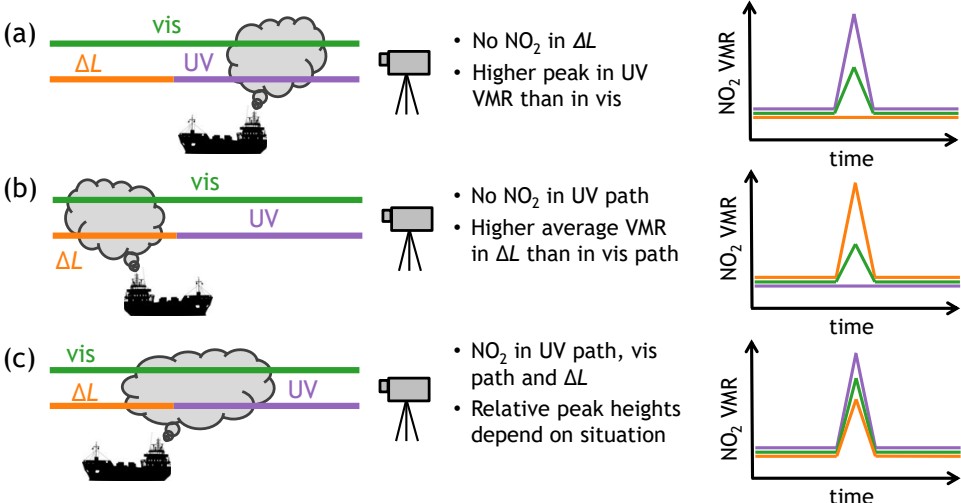

**Figure 2.** Plume–light-path geometry and the resulting path averaged $NO_2$ concentrations for three possible cases: When the plume is close to the instrument and completely covered by the UV path (a), when the plume is further away from the instrument than the UV scattering point and is only covered by the visible path (and $\Delta L$) (b) and when the plume is located around the UV scattering point (c)

and therefore shorter than the complete visible path. On Neuwerk, such a situation can occur for southerly winds (compare Fig. 1b). Section 4.3 shows example measurement results for this kind of situation.

In case (c) the plume is close to the UV scattering point. All three light paths see enhanced $NO_2$. The relative peak heights depend on the fraction of plume $NO_2$ probed by the different light paths as well as the total light paths lengths. On Neuwerk, situations like this will most likely occur for westerly and easterly winds.

As already discussed in Seyler et al. (2017), the measured column density as well as the path-averaged concentration do not only depend on the emitted amount of $NO_2$ inside the plume, but also on the angle of intersection between plume and line of sight of the instrument. The smallest absorptions, and thus column amounts, will be retrieved if the plume runs orthogonally to the line of sight, the highest values if the instrument measures along the plume. The latter can occur for certain combinations of wind direction and speed and ship movement direction and speed. But as the movement of the ship together with the measured wind can result in an apparent wind direction very different from the measured wind direction (Berg et al., 2012), a measurement along the measured wind direction (windward, i.e. pointing anti-parallel to the wind vector) does not in general correspond to measurements along the plume.

The time span between plume emission and measurement is important for the measured $NO_2$ values because of NO to $NO_2$ titration in the plume ($NO + O_3 \rightarrow NO_2 + O_2$), as a large fraction of nitrogen oxides ($NO_x$) is emitted as NO (Alföldy et al., 2013; Zhang et al., 2016), which does not absorb in the spectral range covered and cannot be measured with MAX-DOAS.

Therefore, the $NO_2$ content in the plume is expected to increase with distance from the ship until a steady state is reached. Middleton et al. (2007) modeled the NO to $NO_2$ conversion in plumes at short ranges depending on the $O_3$ concentration. For $O_3$ VMRs of 30 to 50 ppb (20 to 70 ppb), which are typically measured at our Neuwerk station in summer, they predicted the steady state to be reached after 3 to 4 minutes and in the steady state the fraction of $NO_2$ on the overall $NO_x$ to be 65–70 %.

For very fresh plumes shortly after emission, Alföldy et al. (2013) found that the $NO_2$-to-$NO_x$ ratio in the plume does not depend on ambient ozone concentrations, as diffusion limits the availability of $O_3$. Airborne imaging DOAS measurements during an overflight over a ship and its plume from the NOSE campaign on 21 August 2013 presented by Meier (2018) show an increase in $NO_2$ with flown distance from the ship overpass. After the airplane covered a distance of around 3 km the values stabilize and do not increase further. Applying the plume modeling approach discussed in Section 3.3, the plume age at this

point where presumably the steady state was reached was estimated to be around 6.5 minutes, in which the respective plume air parcel traveled a distance of $\sim 1.5$ km. Other, unpublished in situ measurements of ship plumes indicate that after 8–10 minutes at the latest the plume NO content is below 20–30 % for all ships. In view of these findings, as the plumes investigated in our study are mostly older than 10 minutes, we expect and assume the steady state to be already reached.

The lifetime of $NO_2$ is on the order of several hours, but as the time scales investigated here are shorter, we expect the

influence to be small.

## 3.3  Plume trajectories and plume modeling

For a more quantitative treatment of the ship emissions, the exhaust plumes and their movement over time need to be considered. Here, ship plume trajectories have been calculated as simple forward trajectories combined with a Gaussian plume model. On a 10 s time grid, at each time step, each point shaped plume air parcel is moved from its old position to a new

position, which depends on wind direction and speed. Each ship emits a new plume air parcel per time step at the respective ship position, thus creating a chainline-like string of plume air parcels. By starting with an initialization period of 3 hours before the respective measurement time, old plumes from ships that passed by the island before and already left the region of interest can be included in maps as those shown in Fig. 4.

Plume broadening and dispersion over time is accounted for by modeling the width and height of the plumes with a Gaussian

plume model (Pasquill, 1961; Gifford, 1961), an often used model for point source emitters like power plants. It describes the vertical and horizontal plume dispersion with two Gaussian curves and links the pollutant emission rate $Q$, the mean wind speed $U$ (in $x$-direction) and the horizontal and vertical dispersion coefficients $\sigma_y$ and $\sigma_z$ to the concentration $C$ at the point $(x, y, z)$:

$$C(x,y,z) = \frac{Q}{2\pi U \sigma_y \sigma_z} \exp\left(\frac{-y^2}{2\sigma_y^2}\right) \exp\left(\frac{-(z-H)^2}{2\sigma_z^2}\right) \tag{4}$$

Where the vertical coordinate $z$ is corrected for the effective stack height $H$ (the effective height of the plume center line), the sum of the stack height and the initial plume rise.

The dispersion coefficients $\sigma_y$ and $\sigma_z$ are the standard deviations of the Gaussian shaping functions and depend on atmospheric stability. A simple classification scheme defining six different stability classes ranging from very unstable (A) to stable (F) based on wind speed and solar insolation (Pasquill, 1961) is shown in Tab. 2. One set of empirical functions for the dispersion coefficients $\sigma_y$ and $\sigma_z$ as functions of the along wind distance $x$ is given by Martin (1976):

$$\sigma_y(x) = a \cdot x^{0.894} \tag{5}$$

and

$$\sigma_z(x) = c \cdot x^d + f \tag{6}$$

where the distance from the source $x$ is input in kilometers to retrieve $\sigma$ in meters. The stability-dependent empirical constants a, c, d, and f are given in Tab. 3, partially with a distinction between $x \leq 1\,\mathrm{km}$ and $x > 1\,\mathrm{km}$ (Martin, 1976).

As the ships are moving point sources, the course of the plume does not only depend on the wind direction but also on the previous pathway of the ship. The ships move with with a certain direction and speed, thus creating an apparent wind (Berg et al., 2012). It is therefore not sufficient to run the Gaussian plume model for each ship position on each timestep, it has to be combined with the simple forward trajectories, as each plume air parcel has been emitted at a different location. This is done by running the Gaussian plume model for the respective stability class that fits the prevailing weather conditions and creating look-up-tables (LUT) for the plume width and height depending on the distance from the emission point. For each plume air parcel in the trajectory this LUT is then evaluated at the distance this plume parcel traveled since its emission, to retrieve the plume width at this location. The plume width and height LUTs are gained from the Gaussian plume model by going through every $x$ distance in 10 meter steps (a 10 by 10 meter grid is used) and checking in across-wind direction in which distance from the plume centerline the concentration drops under a certain threshold level (in this study: 1/e) compared to the maximum concentration at the plume centerline at this respective $x$ distance. By introducing this kind of normalization, the exact values of the multiplicative factors $Q$ (emission rate) and $U$ (wind speed) become irrelevant for the computation. For the plume width this LUT can be applied to all ships, but for the plume height, as this depends on the stack height, the LUTs have to be computed for each individual ship separately, as their stack heights differ. As neither ship height nor stack height is contained in the broadcasted AIS data, the stack height has to be researched for each ship individually. In this study it was estimated from pictures of a ship by comparing the stack height to the standardized height of the loaded containers. This is not so much of a problem here, as the plume height plays no role for the visual representation of the plumes in the maps (as they represent an aerial view) but only for the detailed analysis of specific plumes of specific ships.

For the method for deriving in plume $NO_2$ VMRs from MAX-DOAS (and airborne imaging DOAS) measurements described in Section 4.4, knowledge of the plume width (and height) is sufficient, so the concentration or emission rate is not modeled here. Plume chemistry like NO to $NO_2$ titration and $NO_2$ loss reactions/$NO_2$ lifetime is neglected. Another source of uncertainty is the fact that the Gaussian plume model only describes an average plume. Each snap-shot in time of a real plume will in general not look like a Gaussian plume, but if multiple snap-shots are averaged over a certain time period, the average shape should approach a Gaussian plume shape. Using the Gaussian plume model for the plume trajectories is therefore only an approximation.

**Table 2.** Atmospheric stability classification scheme (Pasquill, 1961; Turner, 1970) based on surface wind speed and solar insolation: A–very unstable, B–moderately unstable, C–slightly unstable, D–neutral. The additional stability classes E–slightly stable and F–stable occur only at night. For A–B take average of stability parameters (Tab. 3) for A and B.

| Wind speed (10 m AGL) | Solar insolation | | |
|:---:|:---:|:---:|:---:|
| in m/s | Strong | Moderate | Slight |
| < 2 | A | A–B | B |
| 2–3 | A–B | B | C |
| 3–5 | B | B–C | C |
| 5-6 | C | C–D | D |
| > 6 | C | D | D |

**Table 3.** Empirical stability parameters for the computation of the horizontal and vertical dispersion coefficients $\sigma_y$, $\sigma_z$ (Martin, 1976) for the different atmospheric stability classes according to Pasquill (1961). For intermediate stability classes like A–B, averages of parameter values for A and B are taken.

| Stability class | Description | $a$ | $x \leq 1\,\mathrm{km}$ | | | $x > 1\,\mathrm{km}$ | | |
|:---:|:---:|:---:|:---:|:---:|:---:|:---:|:---:|:---:|
| | | | $c$ | $d$ | $f$ | $c$ | $d$ | $f$ |
| A | Very unstable | 213 | 440.8 | 1.941 | 9.27 | 459.7 | 2.094 | -9.6 |
| B | Moderately unstable | 156 | 106.6 | 1.149 | 3.3 | 108.2 | 1.098 | 2.0 |
| C | Slightly unstable | 104 | 61.0 | 0.911 | 0 | 61.0 | 0.911 | 0 |
| D | Neutral | 68 | 33.2 | 0.725 | -1.7 | 44.5 | 0.516 | -13.0 |
| E | Slightly stable | 50.5 | 22.8 | 0.678 | -1.3 | 55.4 | 0.305 | -34.0 |
| F | Stable | 34 | 14.35 | 0.740 | -0.35 | 62.6 | 0.180 | -48.6 |

## 4 Results

### 4.1 Onion peeling approach applied to ship emission measurements

Panel a in Fig. 3 shows the measured $NO_2$ DSCDs in 0.5° elevation for the 335° azimuth direction (compare Fig. 1b) on 26 May 2014. The $NO_2$ shows sharp peaks, which originate from shipping emissions, with rapid changes of $NO_2$ levels between consecutive measurements of up to one order of magnitude. The small, but non-zero baseline between the peaks shows an ambient $NO_2$ pollution, which is enhanced in the morning hours. The background $NO_2$ signal may be originating from land-based sources but may also contain residual, diluted shipping emissions. The morning enhancement might be due to the morning traffic rush hour or boundary layer height changes.

As a result of the longer light path, the $NO_2$ columns measured in the visible range are larger than in the UV. The difference between visible and UV columns, $\Delta$DSCD, shows concurrent peaks for some of the cases, with varying relative height. The peak at 12:50 UTC is not visible in the $\Delta$DSCD, indicating that the plume must be closer to the instrument than the UV scattering point.

Panel (b) in Fig. 3 shows the corresponding effective horizontal light path lengths derived from the measured $O_4$ DSCDs. For a clear sky day like this, path lengths are quite constant over time.

Panel c in Fig. 3 shows the horizontal path averaged $NO_2$ volume mixing ratios retrieved from the $NO_2$ DSCDs by using the $O_4$ scaling approach with the path lengths for UV and visible shown in Panel b, as well as the volume mixing ratio on the path difference calculated via Eq. 3. The baselines of all three curves agree very well, showing that the ambient $NO_2$ background pollution is well-mixed in the boundary layer and homogeneously distributed along all light paths sections. However, the sharp peaks originating from ship emission plumes have different relative heights, showing that the corresponding $NO_2$ field is inhomogeneous. The strong $NO_2$ signal at 12:50 UTC without enhanced $NO_2$ VMR on the path difference, resembling situation (a) in Fig. 2, will be further investigated in the next section.

## 4.2 Northerly wind situations

For northerly winds, the pollution plumes emitted from the ships are blown towards the radar tower, resulting in enhanced $NO_2$ concentrations south of the shipping lane (compare Fig. 1b). In the north of the shipping lane, concentrations should be low, resembling situation (a) in Fig. 2.

In Fig. 4, a 12 minute sequence of consecutive MAX-DOAS measurements on 26 May 2014 starting at 12:46 UTC (14:46 local time) is shown for more detailed investigation of the strong $NO_2$ signal already seen in Fig. 3c at 12:50 UTC. Plotted in each map are the length and location of the UV path and $\Delta L$ as colored lines, with color representing the respective path averaged $NO_2$ VMR. In situ $NO_2$ VMRs are shown as colored dots at the measurement site. Also shown are ship positions and course from AIS data, plume trajectories (see Section 3.3) and wind speed and direction measured by the weather station on Neuwerk.

The sequence of maps shows two ships (magenta triangles) on the shipping lane, moving in opposite directions. The larger ship (length 351 m) moves westward, the smaller ship (length 151 m) moves eastward. The locations of the two plumes (gray shaded stripes) differ considerably due to the different movement directions of the ships and the curved shape of the shipping lane around the island.

For the plume modeling, the stability class C representing slightly unstable conditions has been chosen based on the wind speed and the strong solar insolation on this clear sky day.

In the first panel, the MAX-DOAS measurements at 12:46:24 UTC in 335° azimuth direction are shown. The horizontal path averaged $NO_2$ VMRs are low ($<$ 1 ppb $NO_2$) and agree very well between the different path segments as well as with the in situ measurements, showing that the ambient background $NO_2$ is homogeneously well-mixed in the boundary layer. The fact that the plume from the smaller ship shows up only slightly in the measurements might be due to low emissions from this

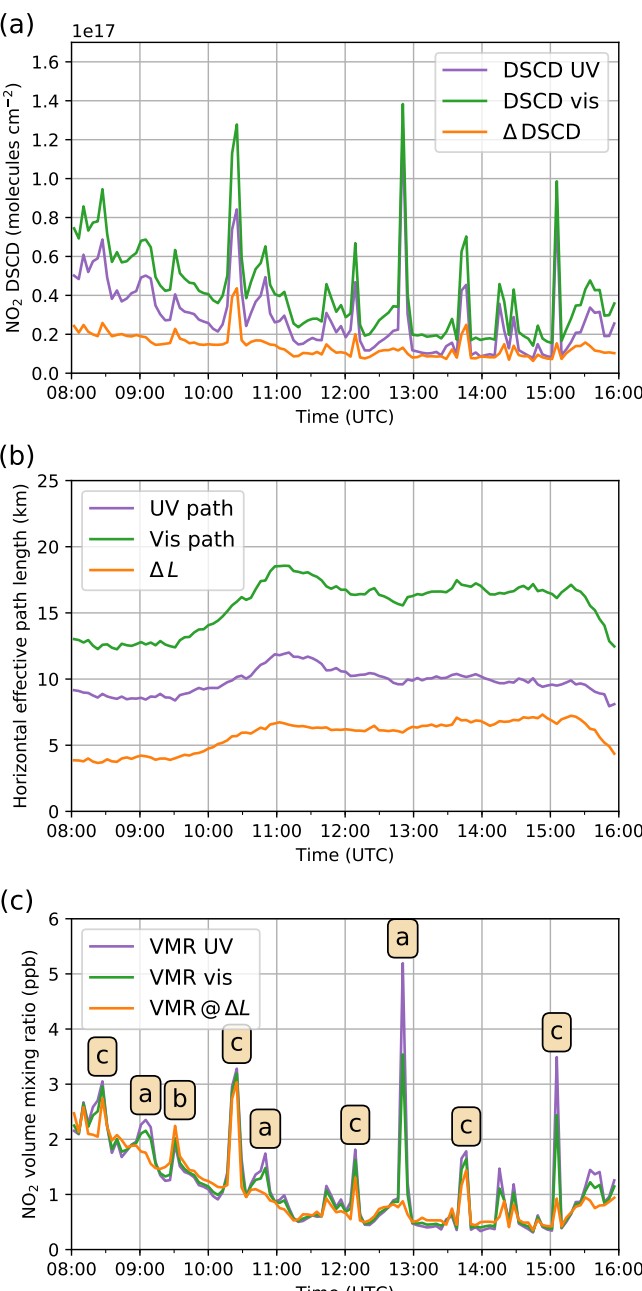

**Figure 3.** Differential slant column densities of $NO_2$ (a), horizontal effective path lengths (b) and horizontal path averaged volume mixing ratios of $NO_2$ on 26 May 2014 in 0.5° elevation and 335° azimuth for the UV (purple) and visible spectral range (green) and their difference (orange)

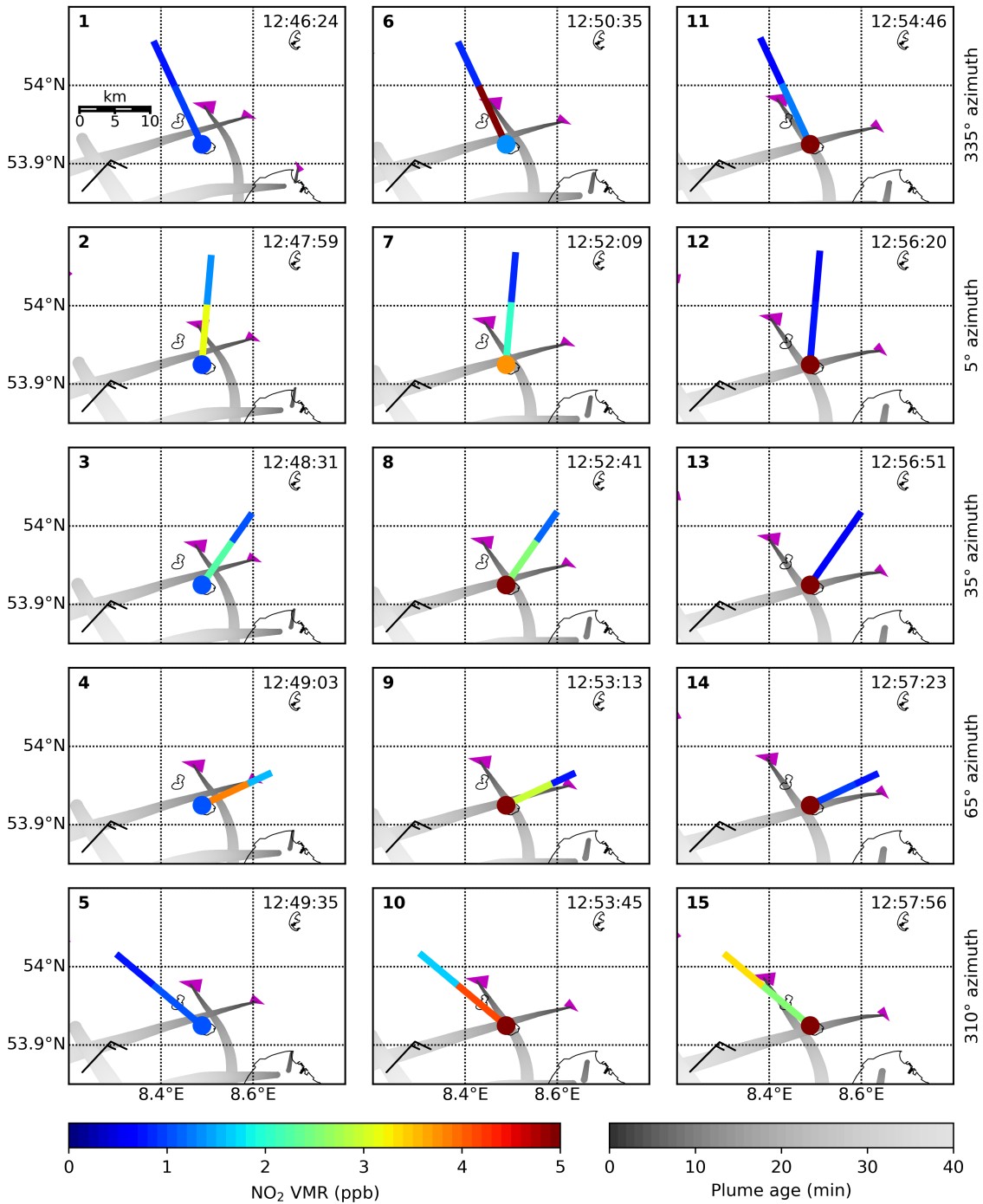

**Figure 4.** Sequence of maps showing 15 consecutive measurements in 0.5° elevation on 26 May 2014, starting at 12:46 UTC (14:46 local time): The extent of the UV path and $\Delta L$ and corresponding path averaged $NO_2$ VMRs are shown as colored lines. In situ $NO_2$ VMRs are shown as a colored dot at the location of the measurement site. Magenta triangles show the ship position and course (sharp tip), with larger triangles for larger ships. The modeled plumes are shown in gray, the lightness of the gray shading representing the plume age. Wind direction and speed is shown with meteorological wind barbs.

comparatively small ship and the dilution of the already strongly dispersed plume, as the plume model predicts a vertical extent of the plume of $\sim 400\,\mathrm{m}$ and a plume width of 1200–1300 m at a plume age of 700-800 seconds.

Panels 2 to 4 (5°, 35° and 65° azimuth, respectively) show enhanced $NO_2$ VMRs (up to 4 ppb) along the UV path close to the instrument, likely due to the plume of the big ship, and low VMRs along $\Delta L$ further away from the instrument. Although MAX-DOAS measurements show enhanced $NO_2$ between site and UV scattering point the plume has not reached the radar tower yet and in situ values therefore stay low.

Panel 5 shows the measurements in the 310° viewing direction which are similar to the measurements in 335° azimuth angle in Panel 1.

In Panels 6 to 10, the plume approaches the radar tower and in situ values begin to rise. MAX-DOAS VMRs are again high close to the radar tower and low in the north of the shipping lane. Due to different angles of intersection between plume and line of sight, the MAX-DOAS path averaged UV VMR is different, showing the highest value of $\sim 5$ ppb when measuring alongside the plume (Panel 6) and much lower values when measuring orthogonally to it (e.g. Panel 3). A small $NO_2$ enhancement of $4 \times 10^{15}\,\mathrm{molec/cm^2}$ is seen in the zenith sky measurements around 12:50 UTC, which is gone at 12:55 UTC, indicating that at least part of the plume was located above the MAX-DOAS instrument. As the zenith sky measurements are used as a sequential reference for the off-axis measurements, this causes a small canceling effect when using the sequential reference. As off-axis DSCDs are on the order of $1 \times 10^{17}\,\mathrm{molec/cm^2}$ reaching up to $1.4 \times 10^{17}\,\mathrm{molec/cm^2}$ as can be seen from Fig. 3, the overall impact on the path averaged VMRs is very small, on the order of 2 to 4 %.

Starting with Panel 9 the in situ instrument measures even higher values, which are not represented in the figure as the color scale extending up to 5 ppb is saturated. In Panel 9 increasing to 6.1 ppb, in Panel 10 and 11 topping at 8.3 ppb and 8.9 ppb, respectively. In Panel 12 the measured $NO_2$ VMR drops to 6.3 ppb but increases again due to the second plume, reaching 6.6 ppb, 6.8 ppb and 7.1 ppb in Panels 13 to 15. After Panel 15, the value increases further to 8.8 ppb and goes down again to ambient background concentrations. This means that the in situ instrument measured two overlapping plumes. The maximum in situ $NO_2$ VMRs are much higher compared to the MAX-DOAS measurements, because the in situ instrument measures directly the $NO_2$ VMR inside the plume and the MAX-DOAS delivers path-averaged values, which underestimate the local VMR inside the plume. The fact that the plume overpass is seen earlier in the MAX-DOAS zenith sky measurements than in the in situ measurements indicates that wind speeds are higher at higher altitudes, so that the upper part of the plume crossed the radar tower earlier than the lower part.

In Panel 11 the ship plume has moved out of the narrow line of sight of the MAX-DOAS instrument and measured $NO_2$ values drop rapidly to ambient concentrations on both path segments. Panels 11 to 14 show all low MAX-DOAS measurements, while the plumes of both ships are hitting the radar tower leading to a very high in situ signal.

In Panel 15 the larger ship has moved further away from the instrument, leading for the first time in this sequence to a higher concentration on $\Delta L$, far away from the instrument, than close by. Comparing the locations of the MAX-DOAS paths with the ship position and modeled plume in detail, however, indicates a much larger intersect of the plume with the UV path than with $\Delta L$. This might be an example probably showing the uncertainty (overestimation) in the path length estimation due to negligence of the correction factor as discussed in Section 3.1.

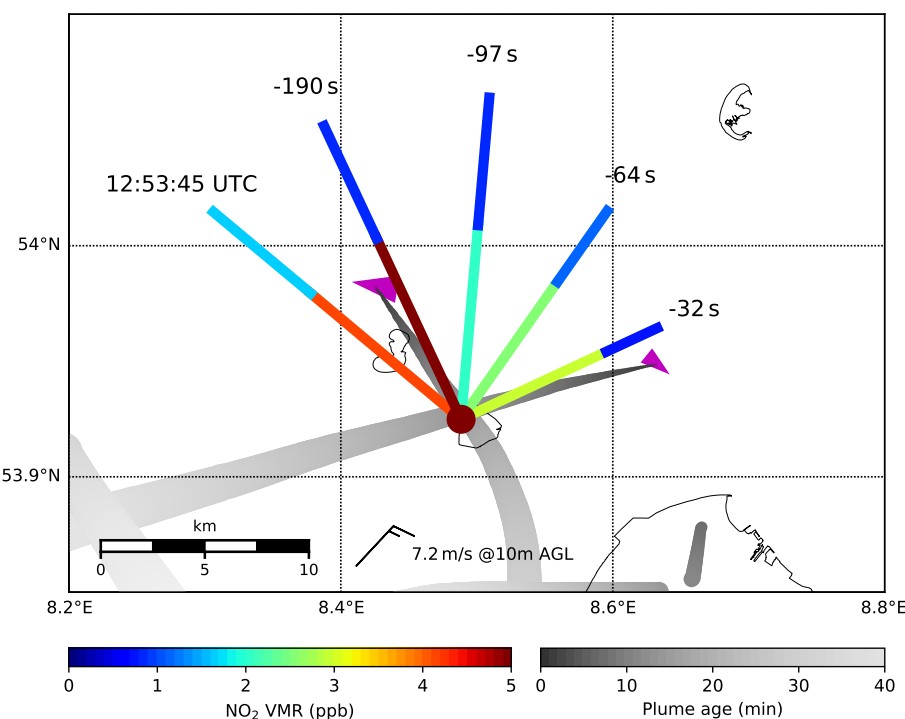

**Figure 5.** Map showing a zoom in onto Panel 10 of Figure 4 and the four previous MAX-DOAS observations, which have been measured between 30 seconds and 3 minutes before the current observation. Horizontal light path lengths (UV path and $\Delta L$) and corresponding path averaged volume mixing ratios of $NO_2$ are shown as colored lines, in situ $NO_2$ values as a colored dot at the location of the instrument. Magenta triangles show the ship position and course, with larger triangles for larger ships. The modeled plumes are shown in gray, the lightness of the gray shading representing the plume age. Please keep in mind that ship and plume position were different for the past measurements. Wind direction and speed is shown with a meteorological wind barb.

Figure 5 shows again, but in more detail the measurements, ship and plume positions from Panel 10. To highlight the entire retrieved two dimensional $NO_2$ field in the measurement region along the shipping lane, the four previous MAX-DOAS measurements are shown as well, which were measured between 30 seconds and 3 minutes before. The strong horizontal gradient between enhanced $NO_2$ concentrations close to the site and low concentrations further away for such a north wind

5  situation is clearly visible in the figure.

### 4.3 Southerly wind situations

The second selected case study shows a diametrically opposite situation: For southerly winds the emitted pollution plumes are blown to the north of the shipping lane (compare Fig. 1b), further away from the instruments. As a result, $NO_2$ concentrations south of the shipping lane, close to the instruments, should be low, resembling situation (b) in Fig. 2. On-site in situ instruments

10  are not able to measure the ship emission plumes.

Figure 6 shows a 12 minute sequence of consecutive measurements on 13 August 2014 starting at 12:35 UTC (14:35 local time). It shows MAX-DOAS path averaged $NO_2$ VMRs as well as in situ measurements. Shown are also ship positions and course from AIS data, plume trajectories (see Section 3.3) and wind speed and direction measured by the weather station on Neuwerk.

In the map sequence, three ships can be seen on the shipping lane, two large ones (336 m and 365 m) and a smaller one (100 m). As all ships move in the same, eastward, direction, the plume trajectories are almost parallel. Apart from the ship emission plumes, another plume crosses the area of interest, originating from the two directly adjacent coal-fired power plants in Wilhelmshaven, located at 53.57°N, 8.14°E, in a distance of about 50 km, southwest of the measurement site. Using the 10 m a.g.l. wind speed of $7.5 \pm 1.0\,\mathrm{m\,s^{-1}}$ the plume age is estimated to be around 110 minutes, and even shorter taking into account that wind speed increases with height.

For the plume modeling, the stability class C representing slightly unstable conditions has been selected based on the wind speed and the strong solar insolation on this clear sky day.

Panel 1 shows the MAX-DOAS measurement at 12:35:31 UTC in the 310° azimuth direction. The horizontal path averaged $NO_2$ VMR along the UV light path is low (~0.6 ppb) and on $\Delta L$ slightly enhanced (~1 ppb), meaning low $NO_2$ close to the instrument and enhanced $NO_2$ further away (than the UV scattering point). The source for the enhanced $NO_2$ signal on $\Delta L$ could either be the small ship's plume or plumes from the more distant power plants.

The next measurement in Panel 2 at 335° azimuth gives similar results. In this viewing direction the plume of the small ship is not in the line of sight of the instrument, indicating that the plume originating from the power plants is the source of the slightly enhanced $NO_2$ VMR along $\Delta L$.

In Panel 3 (5° azimuth) the MAX-DOAS instrument is measuring towards the two adjacent plumes of the two large ships, one located close to the UV scattering point and the other one further away. $NO_2$ VMR is high (~2 ppb) behind the UV scattering point and medium high (~1 ppb) closer to the instrument.

Panel 4 (35° azimuth) shows again high values far away from the instrument and medium high values close by.

In Panel 5 (65° azimuth), only one of the two plumes is in the line of sight and is further away than the UV scattering point, leading to enhanced $NO_2$ along $\Delta L$ and low (ambient) $NO_2$ along the UV path.

Panels 6 and 7 are similar to Panels 1 and 2, showing that the situation in these viewing direction has not changed four minutes later.

In Panel 8, four minutes after Panel 3, the plumes of the two big ships traveled a bit further northward, making the gradient between $NO_2$ VMRs on UV path and $\Delta L$ even stronger.

Panels 10 to 12 are similar to Panels 5 to 7.

In Panels 13 to 15, the plumes of the two big ships are now clearly only probed by the visible light path giving enhanced $NO_2$ concentrations along $\Delta L$ and low, ambient $NO_2$ concentrations along the UV path.

In all 15 consecutive measurements shown in the map sequence the in situ instrument measured constantly low values. This indicates that for southerly winds it cannot detect ship emission plumes at this site. Measured $NO_2$ VMRs agree very well with ambient $NO_2$ VMRs from the MAX-DOAS, retrieved south of the shipping lane along the UV path.

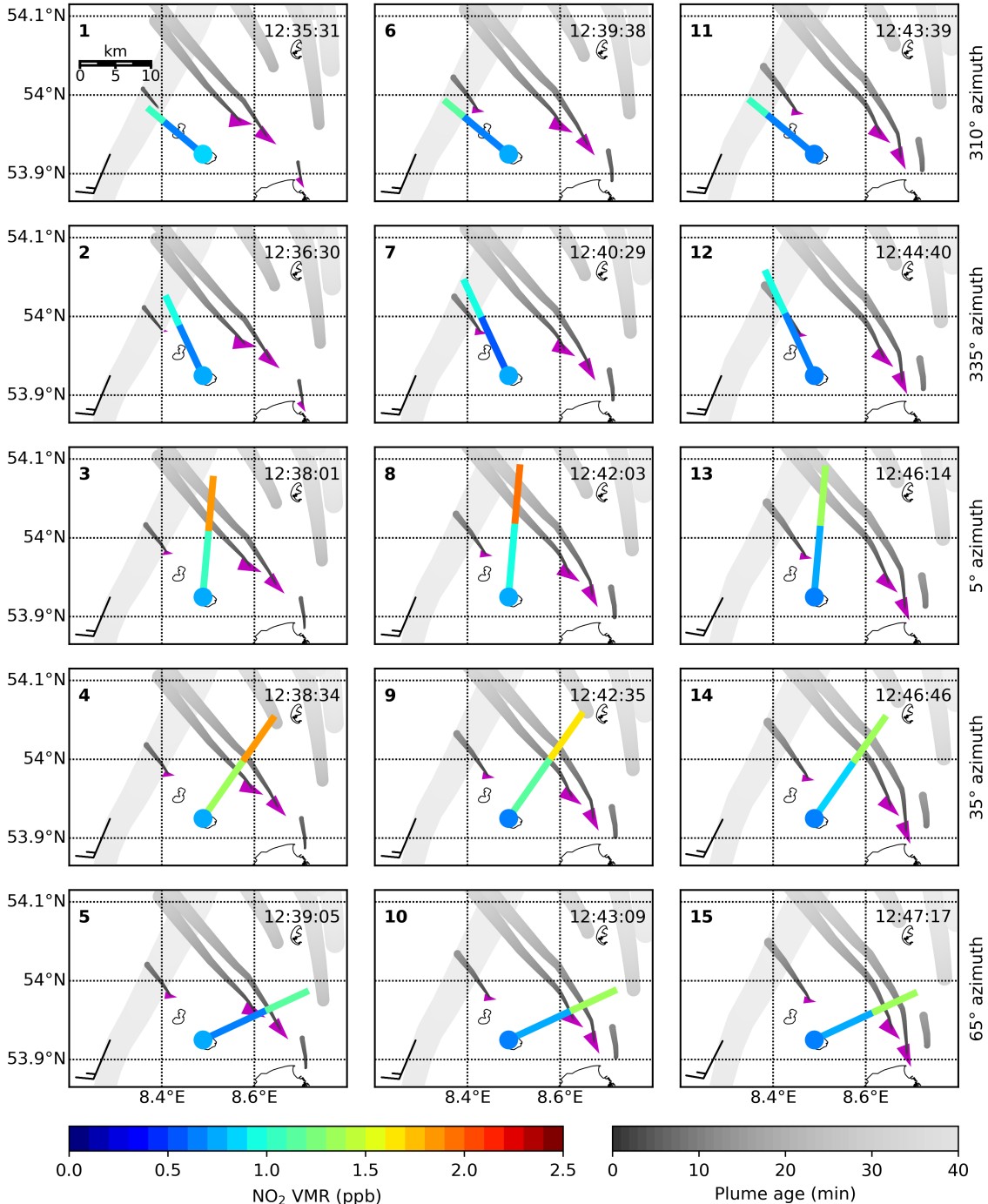

**Figure 6.** Sequence of maps showing 15 consecutive measurements in 0.5° elevation on 13 August 2014, starting at 12:35 UTC (14:35 local time): The extent of the UV path and $\Delta L$ and corresponding path averaged $NO_2$ VMRs are shown as colored lines. In situ $NO_2$ VMRs are shown as a colored dot at the location of the measurement site. Magenta triangles show the ship position and course (sharp tip), with larger triangles for larger ships. The modeled plumes are shown in gray, the lightness of the gray shading representing the plume age. Wind direction and speed is shown with meteorological wind barbs.

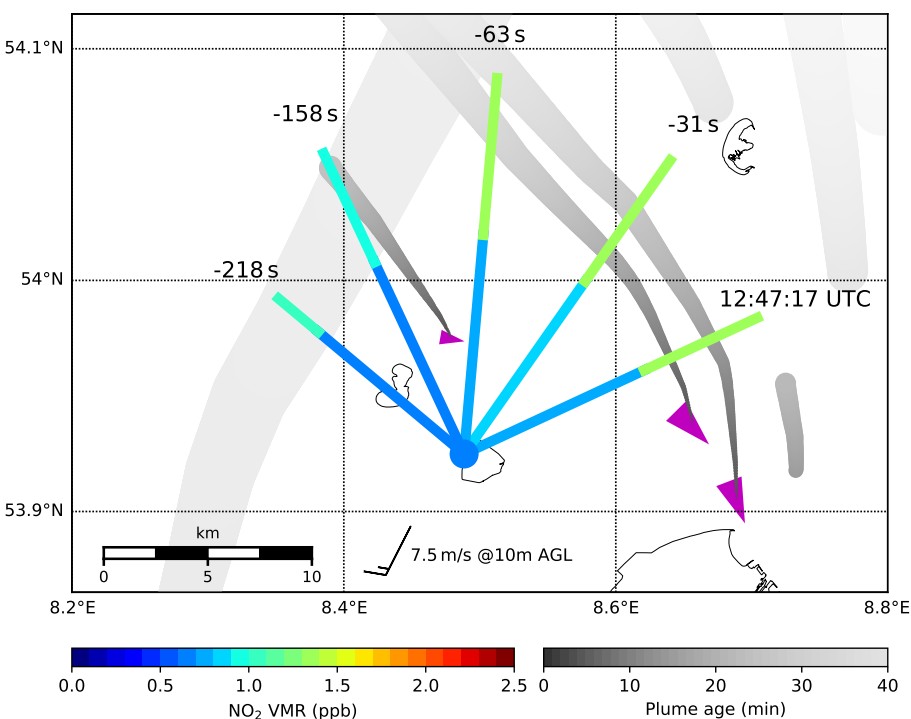

**Figure 7.** Map showing a zoom in onto Panel 15 of Figure 6 and also the four previous MAX-DOAS observations, which have been measured between 30 seconds and 3.5 minutes before the current observation. Horizontal light path lengths and corresponding path averaged volume mixing ratios of $NO_2$ are shown as colored lines, in situ $NO_2$ values as a colored dot at the location of the instrument. Magenta triangles show the ship position and course, with larger triangles for larger ships. The modeled plumes are shown in gray, the lightness of the gray shading representing the plume age. The broader plume in the eastern part of the map originates from the Wilhelmshaven power plants. Please keep in mind that ship and plume position were different for the past measurements. Wind direction and speed is shown with a meteorological wind barb.

Figure 7 shows again in more detail the measurements, ship and plume positions from Panel 15. To highlight the entire retrieved two dimensional $NO_2$ field in the measurement region along the shipping lane, the four previous MAX-DOAS measurements are shown as well, which have been measured between 30 seconds and 3.5 minutes before. It highlights the horizontal gradient between low $NO_2$ concentrations close to the site and enhanced concentrations further away, northward of the shipping lane, demonstrating that with MAX-DOAS it is well feasible to measure ship emission plumes under conditions unfavorable for in situ measurements.

## 4.4 Computation of in-plume $NO_2$ volume mixing ratios using plume modeling and validation with airborne imaging DOAS measurements

In addition to visualizing the two-dimensional $NO_2$ field over the shipping lane, plume modeling allows to retrieve in-plume $NO_2$ VMRs from the MAX-DOAS measurements. For a demonstration of the method, a day was chosen on which simultaneous airborne imaging DOAS measurements were performed, which can be used to validate both the plume modeling and the MAX-DOAS in-plume $NO_2$ VMRs.

### 4.4.1 Computation of in-plume $NO_2$ VMRs

The $O_4$ scaling and onion peeling method yields $NO_2$ VMRs which are averaged along a certain effective horizontal light path. In the general case that the plume does not cover the entire path, the retrieved path-averaged VMR is lower than the in-plume VMR. Thus, to retrieve the in-plume VMR, the fraction of the path probing the plume and thus the plume width has to be known. An estimate for the plume width is provided by the combination of forward trajectory and Gaussian plume model implemented in this study.

Figure 8 shows MAX-DOAS path averaged $NO_2$ VMRs and modeled plumes on 21 August 2013 around 9:53 UTC (11:53 local time). Also shown are AirMAP vertical columns of $NO_2$ which are used for validation in the second part of this section (see Section 4.4.2 for more details).

The Gaussian plume model was run for a stability class of B–C, which was selected due to the moderate insolation (cloudy in the morning, later clearing up) and wind speeds between 3 and 4 meters per second. For this intermediate stability class B–C, representing slightly to moderately unstable conditions, the mean of the parameter values for B and C from Tab. 3 is taken. Wind speed and direction are taken from the weather station on Scharhörn.

Along $\Delta L$ where one of the modeled plumes is located, he MAX-DOAS measured enhanced $NO_2$ compared to the ambient background $NO_2$ measured along the UV path. This plume originates, the one from the $277\,\mathrm{m}$ ship that left the map region to the west. At the intersection of plume and MAX-DOAS line-of-sight, the plume air parcels had traveled a distance of $(2180 \pm 30)\,\mathrm{m}$ in $(660 \pm 10)\,\mathrm{s}$ since emission and the plume model yields a width of $(720 \pm 20)\,\mathrm{m}$. The selection of the stability class clearly has a strong influence on the modeled plume width, as the more unstable class B yields $(870 \pm 20)\,\mathrm{m}$ and the more stable class C yields $(580 \pm 20)\,\mathrm{m}$. This span of values gives a more realistic error estimate. The MAX-DOAS LOS "hits" the plume at an angle of approximately $70°$, so the resulting effective plume width is $(760 \pm 160)\,\mathrm{m}$.

For the computation of the MAX-DOAS average in-plume $NO_2$ VMR, the partial horizontal column inside the plume has to be determined as only scaling the VMR would not account for the background signal. The three panels in Fig. 9 show the MAX-DOAS DSCDs of $NO_2$ for the lowest 5 elevation angles measured in the UV and visible spectral range, as well as their differences, $\Delta$DSCD.

At 9:53 UTC a $\Delta$DSDC of $1.3{\times}10^{16}\,\mathrm{molec\,cm^{-2}}$ is measured along a $2.4\mathrm{km}$ $\Delta L$. The UV measurement of $3.0{\times}10^{16}\,\mathrm{molec\,cm^{-2}}$ along a $7.7\mathrm{km}$ $L_{\mathrm{UV}}$ can be used to estimate the background signal along $\Delta L$. With the modeled plume width $b = (760 \pm 160)\,\mathrm{m} = (76000 \pm 16000)\,\mathrm{cm}$ this yields for the column inside the plume:

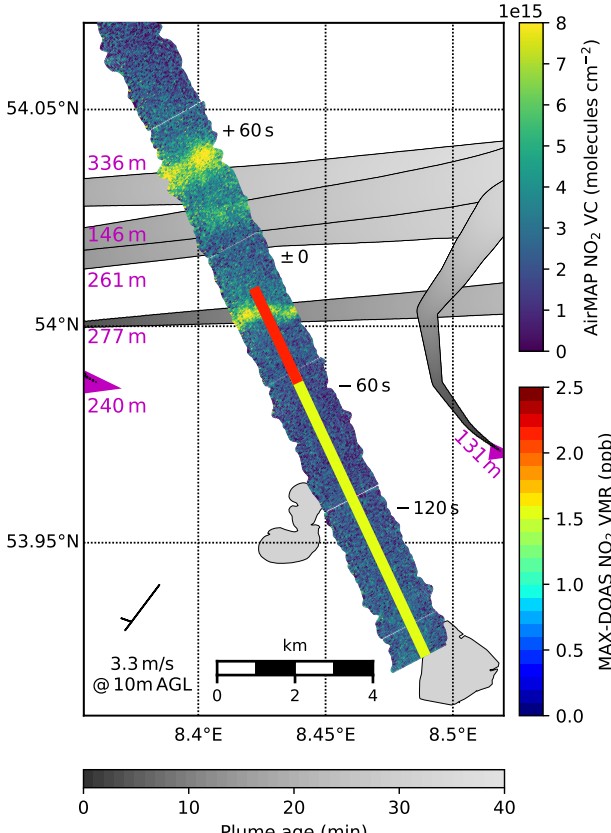

**Figure 8.** Map showing the MAX-DOAS path averaged VMRs (colored lines) and AirMAP vertical columns of $NO_2$ (broad image stripe beneath) on 21 August 2013 around 9:53 UTC (11:53 local time). As the plotted physical quantities are entirely different (VMRs and columns), color scale agreements are not expected (and completely random). Magenta triangles show current ship positions and course, magenta numbers denote the ship length. The modeled plumes (for the MAX-DOAS measurement time) are shown in gray, the lightness of the gray shading representing the plume age. The time difference between AirMAP and MAX-DOAS measurements is indicated in the map at specific parts of the flight track. Wind direction and speed is shown with a meteorological wind barb.

$$\text{DSCD}_{\text{plume}} = \Delta\text{DSCD} - \text{DSCD}_{\text{background}}$$

$$= \Delta\text{DSCD} - \text{DSCD}_{\text{UV}} \cdot \frac{\Delta L - b}{L_{\text{UV}}}$$

$$= (6.9 \pm 3.1) \times 10^{15} \, \text{molec} \, \text{cm}^{-2}$$

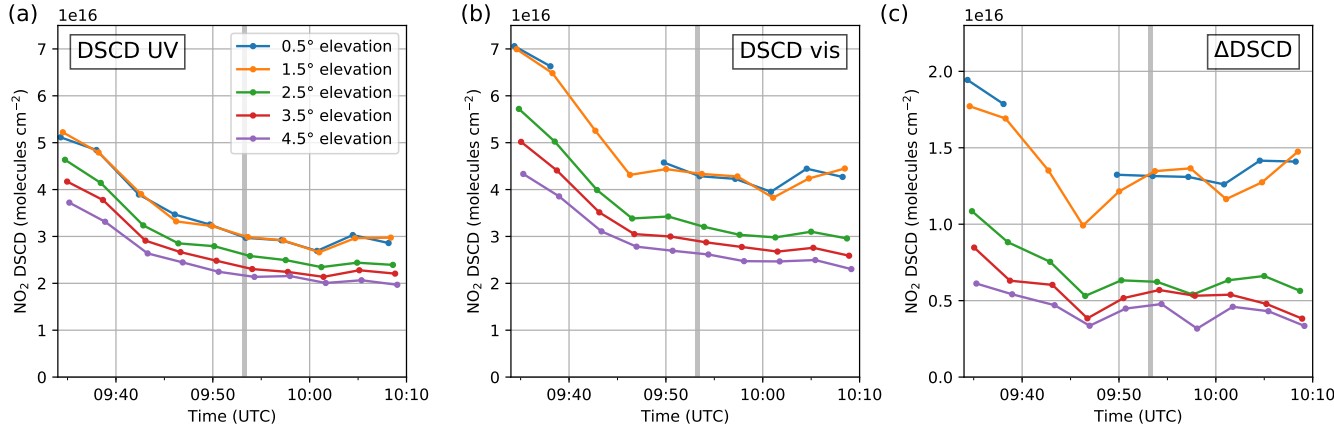

**Figure 9.** MAX-DOAS differential slant column densities of $NO_2$ in the UV (a) and visible (b) spectral range as well as their difference $\Delta$DSDC (c) for the five lowest elevation angles for the azimuthal viewing direction of 335°. The vertical gray line indicates the AirMAP plume overpass time.

where the associated uncertainty has been computed with Gaussian error propagation from the uncertainties of the retrieved DSCDs ($\pm 10\%$), path lengths ($\pm 20\%$) and modeled plume width (see above) assuming independent random uncertainties in the individual variables.

The average VMR inside the plume is given by:

$$\text{VMR}_{\text{plume}} = \frac{\text{DSCD}_{\text{plume}}}{b \cdot n_{\text{air}}} = (3.6 \pm 1.8) \times 10^{-9} = (3.6 \pm 1.8)\,\text{ppb}$$

where $n_{\text{air}} = 2.54 \times 10^{19}\,\text{molec cm}^{-3}$ is the number density of air for the measured pressure of $1025.2\,\text{hPa}$ and temperature of $19.2\,°\text{C}$. The total uncertainty has again been computed with error propagation.

### 4.4.2 Validation

As already indicated above, a comparison to on-site in situ trace gas analyzers is well suited to validate the MAX-DOAS ambient $NO_2$ background values, but fails for in-plume concentrations in many constellations. For unfavorable wind conditions, like southerly winds, the in situ instrument does not detect the plumes at all. The spatial resolution of satellite instruments is not sufficient to resolve individual ship plumes, even with the Sentinel 5 precursor satellite ($3.5 \times 7\,\text{km}^2$, Veefkind et al. 2012).

Airborne imaging DOAS measurements, as have been performed in the region of interest during the NOSE (for german "Nord-Ost-See-Experiment" meaning "North and Baltic sea experiment") campaign (Meier, 2018) on 21 August 2013, are the ideal method for validation of our results. Mapping of the MAX-DOAS line-of-sights, as has been done during NOSE, allows to compare the approximate plume position retrieved from the onion peeling MAX-DOAS method and those from the plume modeling to the real plume position.

Delivering high resolution $NO_2$ maps of the plumes, the airborne measurements can be used to validate both the plume positions calculated with simple forward trajectories and the plume width retrieved from the Gaussian plume model. By incorporating plume height information from either plume modeling or the vertical elevation scans of the MAX-DOAS, an average in-plume $NO_2$ VMR can be computed from the airborne vertical column measurements and compared to the result from the MAX-DOAS.

The *Airborne imaging Differential Optical Absorption Spectroscopy instrument for Measurements of Atmospheric Pollution* (AirMAP), being installed on a Cessna research aircraft of the Freie Universität Berlin for the measurements, is a push-broom imaging DOAS instrument. Scattered sunlight from below the aircraft is collected by a wide-angle objective and coupled into a bundle of 35 sorted optical fibers. The image of the vertically stacked fibers is then dispersed by an imaging grating spectrometer and mapped onto a frame-transfer-CCD. The total field of view of around 52° leads to a ground swath width similar to the flight altitude. With this set-up, 35 across track pixels are measured simultaneously with an exposure time of 0.5 seconds, leading to a spatial resolution better than 50 m when the aircraft is flying at 1600 m altitude. For more detailed information on the instrument see Schönhardt et al. (2015) and Meier et al. (2017).

In the AirMAP data analysis, differential slant column densities of $NO_2$ were retrieved in a fit window of 425-450 nm using the settings described in Meier et al. (2017). For the retrieval of $NO_2$ vertical column densities, air mass factors were calculated for an $NO_2$ box profile assuming constant $NO_2$ in the lowest 500 m, in an atmosphere without aerosols and for a constant surface reflectance of 0.05. This box profile height is an educated guess on an upper limit for the typical vertical plume extent for older ship plumes, which the plume modeling has proven to be in the right order of magnitude.

Fig. 8, already mentioned in the previous subsection, shows additionally to the MAX-DOAS path averaged VMRs the AirMAP vertical columns of $NO_2$ for a ship plume measured on 21 August 2013 around 9:53 UTC (11:53 local time). At about that time, the aircraft flew along the MAX-DOAS 335° azimuth line-of-sight crossing the shipping lane and mapping multiple ship plumes. Enhanced $NO_2$ is measured where the aircraft overpassed the plumes, revealing the location and horizontal extent of the plumes. The southernmost plume was also covered by the MAX-DOAS instrument's $NO_2$ measurement in the visible spectral range. As a result, the path averaged $NO_2$ VMR along $\Delta L$ shows enhanced values compared to the ambient background $NO_2$ measured along the UV path, indicating a plume somewhere along $\Delta L$, which is validated by the airborne measurements. Along the UV path AirMAP $NO_2$ VCDs are significantly lower confirming the assumption of ambient background pollution.

The time difference between both measurements of less than 20 seconds is very small, especially considering the integration time of the MAX-DOAS instrument of 10 seconds. The position of the plume (calculated with a forward trajectory) and the horizontal extent of the plume (computed with the Gaussian plume model) matches the real plume positions measured by AirMAP very well. The plumes further north have been measured by AirMAP around 1 minute later, enough time for the wind to blow the plumes northward so that the positions do not fully coincide with the plume forward trajectories which have been computed for the MAX-DOAS measurement time. Inspecting the AirMAP measurements in detail reveals that the real plumes are not as smooth as the modeled plumes and show some irregularities and random fluctuations caused by turbulence. This deviation is expected, as the Gaussian plume model used here assumes a steady state and describes a (long) time averaged

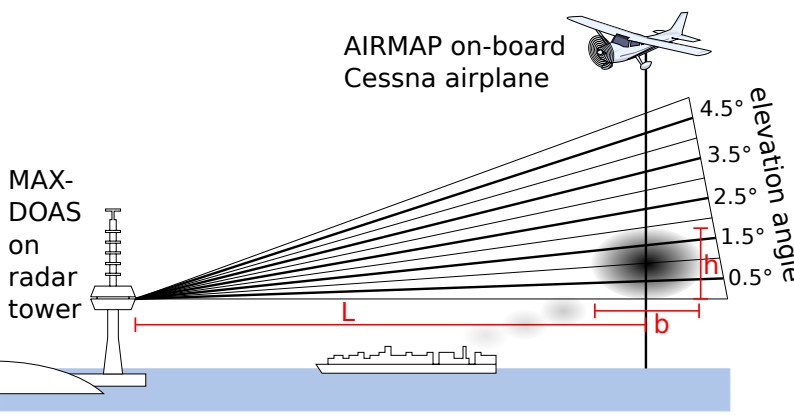

**Figure 10.** Sketch of the different measurement geometries of ground-based MAX-DOAS and airborne imaging DOAS instrument when measuring a ship plume. While the MAX-DOAS instrument scans the plume vertically, the AirMAP instrument measures in nadir direction. Distances, heights and sizes are not to scale.

picture of a plume. Nevertheless, the modeled plume widths fit quite well. These results provide confidence in the modeled plume trajectories, as well as in the onion peeling approach to detect locally enhanced $NO_2$ levels in the $\Delta L$ light path segment.

For the validation of the in-plume $NO_2$ VMR by AirMAP, one has to consider the crucial differences in viewing geometries which are sketched in Fig. 10. The MAX-DOAS instrument measures (slightly slanted) horizontal transects of the plume and can scan the plume vertically by using different elevation angles. The AirMAP instrument, measuring in nadir direction downward from the aircraft, observes vertical transects of the plume. The AirMAP measurements deliver vertical columns of $NO_2$ between ground and aircraft, but no information about the vertical location of the $NO_2$ inside the column. By assuming a box profile for the near-ground $NO_2$ layer (the plume), one can derive mixing ratios from the vertical columns, but for this the vertical extent of the plume has to be known. This plume height $h$ can either be taken from the plume modeling or can roughly be estimated from the MAX-DOAS vertical scan measurements if the distance to the plume is known, as it is from the airborne measurements.

The Gaussian plume model delivers a height of $(320\pm20)\,\mathrm{m}$ reaching from the ground to this height at the respective distance from the emission point (see above) for the selected stability class B–C and for a stack height of $40\,\mathrm{m}$ estimated from pictures of the ship and an assumed initial plume rise of $10\,\mathrm{m}$. Again, the selection of the stability class has an influence on the modeled plume height, with $(420\pm20)\,\mathrm{m}$ retrieved for the more unstable class B and $(230\pm20)\,\mathrm{m}$ for the more stable class C, the span giving again an idea on the uncertainty introduced by the selection of the stability class.

For the estimation from the MAX-DOAS measurements, we need to reconsider Fig. 9, showing the MAX-DOAS DSCDs of $NO_2$ for UV and visible spectral range as well as the $\Delta$DSCD. The UV measurements in Panel (a) show the typical elevation angle dependency for tropospheric absorbers, with longest light paths and therefore highest DSCDs in the lowest elevation

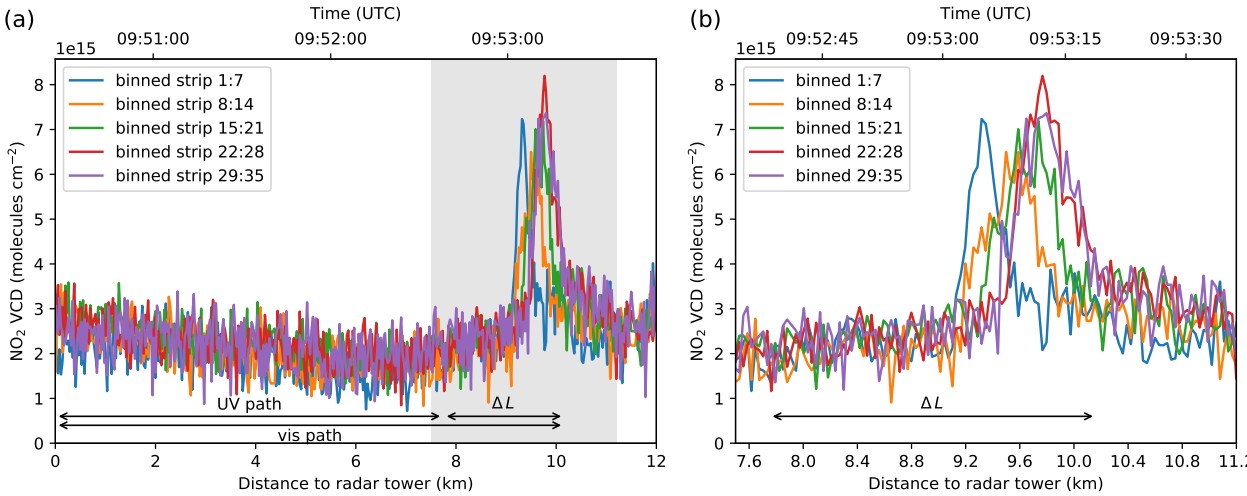

**Figure 11.** AirMAP vertical columns of $NO_2$ as a function of distance (lower axis) or time (upper axis) for the flight track section shown in Fig. 8. The right plot is a zoom in on the gray shaded area. Horizontal arrows denote the horizontal effective light paths.

angles. When the instrument points further up (i.e. higher elevation angles), the light path length through the troposphere decreases giving smaller DSCDs.

Comparing Panel (b), showing the visible measurements, to Panel (a), the values are in general larger due to the longer light path length for longer wavelengths but show a similar separation except for the "gap" between the low elevations (0.5°, 1.5°) and higher elevations (2.5°, 3.5°, 4.5°). This implies that there is even more additional $NO_2$ in the lower elevations than is expected from the longer light path effect. The gap is even more pronounced when inspecting the $\Delta$DSCD shown in Panel (c). This excess $NO_2$ most certainly originates from the ship emission plume. Assuming that the plume vertically fills the whole vertical field of view of the 0.5° and 1.5° elevation an upper boundary for the plume height $h$ can be calculated. The field of view of the instrument is around 1.0°. Thus the plume is observed in a solid angle of 2.0° (compare Fig. 10). At a distance of 9.6 km (see below), this corresponds to a plume height of $h = 9.6\,\text{km} \cdot \tan 2° \approx 335\,\text{m}$. This result is in good agreement with the plume modeling result of $(320 \pm 90)\,\text{m}$.

Figure 11a shows $NO_2$ VCDs from AirMAP as a function of distance to the radar tower for the flight track section shown in Fig. 8. The 35 individual viewing directions were binned to 5 (1:7, 8:14, 15:21, 22:28 and 29:35) to reduce the noise. Although additional binning would reduce the noise even further, it would also smear out the plume signal, since the flight track crosses the plume not orthogonally but at an angle of about 70° (see Fig. 8). A strong enhancement of $NO_2$ is observed at a distance of about 9.1 km to 10.1 km, as it was expected from the MAX-DOAS $NO_2$ enhancement along $\Delta L$. Figure 11b shows the measurements of the plume in more detail, revealing the distance shift of the plume position in the different AirMAP viewing directions due to the slanted angle between flight direction and plume. The $NO_2$ enhancement caused by the plume is roughly

Gaussian-shaped in all 5 binned viewing directions, confirming that the Gaussian plume model gives a good approximation of the plume shape, although maximum values and peak widths differ due to the turbulent fluctuations.

The measured vertical columns are total columns between flight altitude and ground level. To retrieve the local enhancement of $NO_2$ inside the plume, the estimated background column containing ambient $NO_2$ is subtracted from the total $NO_2$ column:

$$VC_{plume} = VC_{total} - VC_{background}$$
$$= (7.0 \pm 2.0) \times 10^{15}\,\mathrm{molec\,cm^{-2}} - (3.2 \pm 1.0) \times 10^{15}\,\mathrm{molec\,cm^{-2}}$$
$$= (3.8 \pm 2.2) \times 10^{15}\,\mathrm{molec\,cm^{-2}}$$

Possible error sources for the AirMAP measurements are fitting uncertainties on the retrieved DSCDs, uncertainties on the surface reflectance, the assumed profile shape and aerosols, while uncertainties on the $NO_2$ amount in the reference spectrum

cancel out when subtracting the background, yielding a maximum overall uncertainty on the $NO_2$ VCD of about $30\,\%$ (Meier et al., 2017).

The $NO_2$ columns measured horizontally (MAX-DOAS) and vertically (AirMAP) through the plume are different. This is expected, because the horizontal and vertical extent of the plume differ – the plume width being approximately two times larger than its height. For a quantitative comparison, the in-plume $NO_2$ column density needs to be converted to an average in-plume

VMR.

$$VMR_{plume} = \frac{VC_{plume}}{h \cdot n_{air}}$$

yielding $(4.7 \pm 3.0)\,\mathrm{ppb}$ for $h = 320\,\mathrm{m}$ (from plume model) or $(4.5 \pm 2.7)\,\mathrm{ppb}$ for $h = 335\,\mathrm{m}$ (from MAX-DOAS), where the overall uncertainty has been computed with error propagation.

This result is in reasonably good agreement with the average in-plume VMR of $(3.6 \pm 1.8)\,\mathrm{ppb}$ derived from the MAX-DOAS

measurements combined with the Gaussian plume model. Having the AirMAP measurements for validation, the plume width computed by the plume model can be compared to the AirMAP measurements (Fig. 11). Using the same threshold as for the modeling, 1/e, this gives a plume width $b$ of 600–700 m, or an effective plume width of $b_{eff} = (690 \pm 53)\,\mathrm{m}$ due to the 70° angle between plume and flight direction. These values are in good agreement with the slightly higher values of $b = (720 \pm 150)\,\mathrm{m}$ and $b_{eff} = (760 \pm 160)\,\mathrm{m}$, respectively, computed by the model, again confirming the validity of the Gaussian plume modeling

approach for this study. Using the more accurate plume width estimate from the AirMAP measurements, the MAX-DOAS in-plume VMR changes to a value of $(4.0 \pm 1.8)\,\mathrm{ppb}$ giving an even better agreement with the AirMAP results. A thorough inspection of the AirMAP measurements along the UV path of the MAX-DOAS (see Fig. 8) reveals that there is a slight decrease of ambient $NO_2$ background pollution observed along the UV path from the radar tower towards the UV scattering point and towards the plume location. Estimating the background column along $\Delta L$ from the UV path, where the MAX-DOAS

delivers only one averaged value, thus might lead to a small bias in the background correction. If too much $NO_2$ is subtracted, the MAX-DOAS in-plume DSCD and VMR might be underestimated, which could explain the lower MAX-DOAS value compared to the AirMAP result.

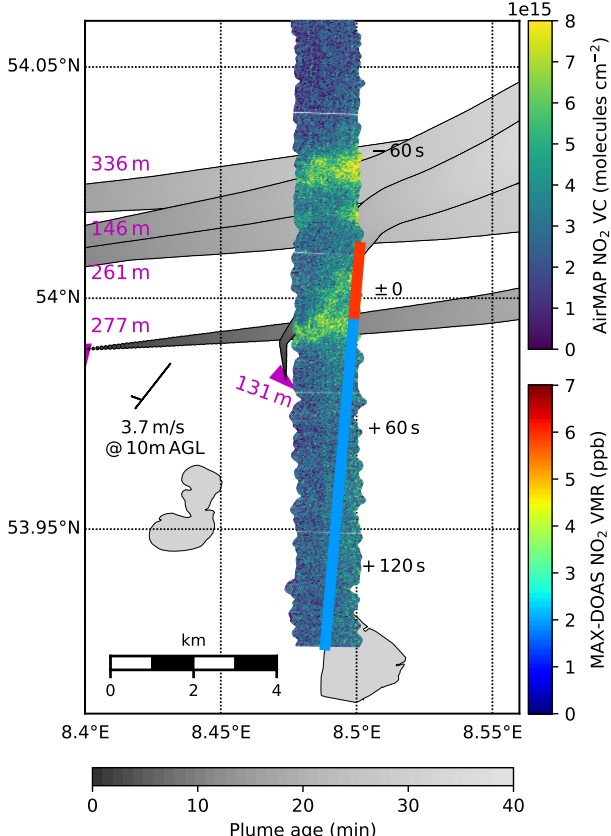

**Figure 12.** Map showing the MAX-DOAS path averaged VMRs (colored lines) and AirMAP vertical columns of NO$_2$ (broad image stripe beneath) on 21 August 2013 around 9:43 UTC (11:43 local time). Magenta triangles show current ship positions and course, magenta numbers denote the ship length. The modeled plumes (for the MAX-DOAS measurement time) are shown in gray, the lightness of the gray shading representing the plume age. The time difference between AirMAP and MAX-DOAS measurements is indicated in the map at specific parts of the flight track. Wind direction and speed is shown with a meteorological wind barb.

Another possible explanation for the lower MAX-DOAS values could be the underestimation of the VMR due to overestimation of path lengths because of negligence of correction factors as mentioned in Section 3.1.

Figure 12 presents another AirMAP overpass over several plumes from ten minutes earlier, again showing good agreement between the measured plume position and the approximate plume positions derived from the onion peeling MAX-DOAS. It shows even better how modeled plumes and real plume positions measured by AirMAP fit together. A computation of in-plume VMRs is not possible in this case, as two plumes are located along $\Delta L$ and they are also not fully covered by $\Delta L$.

## 5 Conclusions

The present study describes a novel application of the "onion-peeling" MAX-DOAS approach to measurements of shipping emissions to estimate the two-dimensional pollutant distribution in the strongly inhomogeneous $NO_2$ field over a shipping lane. The ability to probe air masses at different horizontal distances to the instrument to derive the approximate ship plume positions in the measurement area is shown on the basis of selected case studies out of the three year measurement period on the island Neuwerk. Located in the German Bight, 6–7 km south of the main shipping lane from the North sea into the river Elbe, the island was selected as an ideal site for the onion peeling MAX-DOAS approach as it is in a suitable distance to the shipping lane for exploiting the use of UV and visible radiation to probe the emission plumes released from the passing ships.

To determine the horizontal light path lengths for the onion peeling, a simple approach using the trace gas column of the oxygen collision complex, $O_4$ has been applied. To compare the measurements on the shorter UV path with the measurements on the longer visible path, path-averaged volume mixing ratios have been derived from the measured column amounts of $NO_2$. For the "onion peeling", a separate $NO_2$ VMR along the path difference, usually located over or close to the shipping lane several kilometers away from the instrument, has been computed from UV and visible measurements, allowing to compare $NO_2$ values close to the instrument (along the UV path) and several kilometers away (along the path difference).

It is shown that for northerly wind directions, the onion peeling MAX-DOAS can detect enhanced $NO_2$ concentrations close to the instrument south of the shipping lane and low $NO_2$ concentrations north of the shipping lane. For southerly wind directions, low $NO_2$ values are measured close to the site south of the shipping lane and enhanced $NO_2$ values in the north of the shipping lane, demonstrating that the MAX-DOAS instrument can detect pollution several kilometers away from the instrument under wind directions unfavorable for in situ measurements.

A combination of simple forward trajectories and a Gaussian plume model has been implemented to model the ship plumes, allowing to compute in-plume $NO_2$ volume mixing ratios from the MAX-DOAS measurements, which is demonstrated examplarily for a plume measured on 21 August 2013.

For validation of both the plume modeling and the MAX-DOAS results, airborne imaging DOAS measurements taken by the AirMAP instrument during the NOSE campaign on this very same day have been used. AirMAP's measured plume positions agree well with the ones estimated by using the onion peeling MAX-DOAS approach showing that MAX-DOAS measurements can be used to derive the approximate position of ship emission plumes. The good agreement of modeled plume positions and shapes with AirMAP measurements shows that simple forward trajectories combined with a Gaussian plume model look-up-table approach provide sufficient accuracy to model the two-dimensional $NO_2$ field over the shipping lane.

By incorporating information about the vertical plume extent from either plume model or MAX-DOAS vertical scan measurements, an in-plume $NO_2$ VMR has been derived from AirMAP measurements, too. AirMAP and MAX-DOAS in-plume VMRs agree well within their error margins, confirming the validity of the onion peeling MAX-DOAS approach and the presented method to derive in-plume $NO_2$ VMRs from MAX-DOAS measurements.

To conclude, the presented measurements provide a real world demonstration that the onion peeling approach works for MAX-DOAS measurements and can successfully be applied to investigate air pollution by ships and to derive in-plume $NO_2$ volume mixing ratios for ships passing the instrument in a distance of several km.

*Data availability.* The data used in this study are available from the cited references and directly from the authors upon request.

5 *Competing interests.* The authors declare that they have no conflict of interest.

*Acknowledgements.* The research project which facilitated the reported study was funded in part by the German Federal Maritime and Hydrographic Agency (Bundesamt für Seeschifffahrt und Hydrographie, BSH) and the University of Bremen. The authors thank the Waterways and Shipping Office Cuxhaven (Wasser- und Schifffahrtsamt, WSA), the Hamburg Port Authority (HPA), the AirMAP and NOSE teams and the FU Berlin for their help and support. Many thanks to the editor, Michel Van Roozendael, and to the two anonymous referees for their
10 valuable comments and suggestions, which helped to improve this publication.

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
