# Peer review of "Studies of the horizontal inhomogeneities in NO2 concentrations above a shipping lane using ground-based MAX-DOAS measurements and validation with airborne imaging DOAS measurements"

_Atmospheric Measurement Techniques, 2018_

## Referee Comment (RC1) · Anonymous Referee #1 · 12 Dec 2018

GENERAL COMMENTS

The manuscript "Studies of the horizontal inhomogeneities in NO2 concentrations above a shipping lane using ground-based MAX-DOAS and airborne imaging DOAS measurements" presents nicely and picturesque the onion peeling approach applied to measurements in the German bight, demonstrated on individual measurements. In the second part of the manuscript, the authors compare 2 specific measurement instances to airborne imaging DOAS measurements taken during the NOSE campaign. These two instances show well the validity of the onion peeling approach, qualitatively and

quantitatively.

While this second part, the comparison with imaging doas, makes also quantitative estimates, the first part, showing two times example measurements within approximately 12 minutes in 5 different azimuth directions, stays very qualitative. It neither includes an estimation of errors by e.g. the negligence of the correction factor in the O4 scaling approach for the effective light path estimation, nor does it include an attempt of making use of some ancillary information about the plume using e.g. the STEM model (Jalkanen et al. 2009 acp 9209-9223 , 2012 acp 2641-2659) and some diffusion model to estimate plume width/ height.

The authors argue that the presented method is suited to measure concentrations when wind conditions are unsuitable for surface measurements which do not measure any enhanced concentration if the wind blows the plume away from the measurement station. However, they themselves mention that the measured concentration with the onion peeling approach is not representative of the in-plume concentration. The authors lack to investigate possibilities how to extract useful information using other available information (using more info from the AIS data in combination with the STEM model and better modelling of the plumes). No attempt was made to connect the measured concentration to the in-plume concentration in the first part of the paper. Even if this is not carried out, I strongly recommend the authors to think about ways how this could be done and at least describe what could be done. Without it, this method seems rather incomplete and its usefulness quite limited.

However, the paper is very well presented and shows that if complementary measurements are available, useful estimates about plume concentrations of individual plumes can be calculated and hence should be published! It remains nevertheless unclear what the main purpose of the measurements is if no connection can be made to the in-plume concentration. This could certainly be clarified better both in the introduction and in the conclusions. I recommend to either include attempts to extract more information about the actual plume concentrations by using more ancillary information in

the first part of the paper (basically describing Fig. 6, 7, 8 and 9) or skipping that part and only concentrating on the second part. In the current state, it is a long paper that, over big parts, is rather qualitative and does not give much quantitative information about plume concentrations and hence is not quite suited for measurements of ship emissions on its own.

SPECIFIC COMMENTS

(a) The authors only mention the restrictions on sulphur emission by the MARPOL convention. However, this manuscript is about NO2, so it should probably also give the info on this. Something like... EU adaption of this in form of directive 2012/33/EU NOx emissions depends on the rated rotational speed of the engine crankshaft, implementation in 3 tiers, last one not yet implemented, shifted to ~2021

(b) In the last paragraph of Sect. 3.2, the authors mention the importance of NO to NO2 conversion. Maybe the authors can give some estimates on time- (and spatial) scales for the increase (probably depending on some "standard"(?) background O3 concentration). Also, the authors mention in Sect. 3.3 that plume broadening and dilution over time is neglected. Maybe an order of magnitude estimation should be included and it should be outlined how this information can be used to extract more useful information from the measurements. Which of the two effects dominates for which time-scales? Maybe a reference to some dispersion models that include chemistry?

(c) In Sect. 3.3 it is stated that the initialization period is 90 minutes before the measurements. The big plume (roughly N-S direction) present in all panels in Figure 8 (and 9), originates, according to the authors (Sect. 4.3 3rd paragraph) from two coal-fired power plants in Wilhelmshaven, 50 km away. The authors estimate the plume age to be 110 minutes. However, this suggests that the initialization period needed to be larger than 90 minutes, otherwise the plume would not have had the time to travel that far. I think this should be clarified.

(d) Regarding the plume trajectories, maybe the "apparent wind" approach as illus-

trated e.g. in Berg et al (2012, amt 1085-1098) should be referenced.

(e) It is mentioned that there are two stations measuring wind conditions, one on Neuwerk, one on Scharhörn. However, it is not clear whether the wind used for the calculation of the plume trajectories in e.g. Fig. 6 or Fig.8 is a simple average of the two, or if it depends on the position of the plume at every given moment which wind (some sort of spatial interpolation) is applied, or if only one is used. Please clarify.

(f) Regarding the author's comment to Fig. 6 panel 1 why the plume of the small ship is not seen: Maybe the authors could do a quick calculation which heights are seen at the expected distance of the plume (about from about 20–40 m ?). What do the authors find more likely? Maybe using the STEAM model for that particular ship, together with estimates for the dilution due to diffusion and NO to NO2 conversion the authors could approximate in plume concentration and exclude or not exclude their first alternative.

(g) Can the authors comment on the effect on the MAXDOAS results when the plume is over the instrument, as also indicated by high in-situ measurements? Does this lead to cancelling effects or is the vertical extend of the plume negligible comparable to the horizontal?

(h) Fig6, panel 10: The plume from the big ship cannot have yet reached the VIS-only region (Delta L). However, compared to the measurement 4 minutes before, the VMR seems to have increased by around 1.5 ppb. Any suggestion why?

(i) Similarly, panel 15 seem to indicate a larger intersect of the plume with the viewing direction for the UV region than for the VIS-only region. Still it looks like (as mentioned below, maybe not the best choice of colour map) the VIS-only region has a much higher average VMR. Probably this is due to the effect of overestimated length (due to negligence of correction factor as mentioned by the authors) and hence more of the intersection is in the UV-only path?

(j) The first two (not numbered) equations seem to indicate that the air density in fact

cancels out in the authors approach to estimate the VMR since only surface values for concentrations?

(k) The title suggests a more "equal weight" between the two methods in terms of "being presented". However, the imaging approach seems to be merely used for validation and is not presented as such, since this is done in a different publication. Maybe the title should reflect this.

(l) The authors conclude in their last sentence of the manuscript that this approach can be successfully applied to ship emission measurements. Nowhere in the paper is an estimation of the ship emission presented. I advice the authors to delete or reformulate this sentence.

TECHNICAL CORRECTIONS AND SUGGESTIONS

(a) page 2, line 31: ... is of (not on) the order of....

(b) page 5/6: 2 equations on these sides are not labelled. I think all equations should be labelled.

(c) page 8, last sentence of penultimate paragraph: This is a really confusing sentence. I would probably reformulate to something like: "The movement of the ship together with the measured wind results in an apparent wind direction very different from the measured wind direction. Therefore, a measurement along the measured wind direction does not in genereal correspond to measurements along the plume".

(d) Fig. 3,4 and 5: For easier reference, it might be a good idea to bundle those into one figure with 3 vertically aligned panels.

(e) Fig 6 and 8: maybe a length scale would be nice to include. Also, I suggest to label the viewing directions on the right-hand side on each row. I am not sure if a jet-like colour scale is the best choice. The gnuplot type one used in Fig. 1 or viridis or any other colour scale that is monotone in lightness (The first and the last comment also hold for Fig. 7 and 9) would be better. However, maybe that is just something the

authors can keep in mind for the next publication.

(f) page 13, line 2: "lightboth"??

(g) page 14, Sect. 4.3, second paragraph: Figure 8... (not Figure 6)

(h) Figures 1,6,7,8, and 9: Can the authors quickly state why Nigelhörn and Scharhörn got merged into one island in their map?
* * *

---

## Referee Comment (RC2) · Anonymous Referee #2 · 4 Jan 2019

This paper presents measurements of NO2 from ship emissions in the German Bight using MAX-DOAS instrument, and shows that horizontal information on the NO2 distribution can be derived using an onion peeling method with NO2 slant columns derived separately in the UV and visible, which are observing slightly different air masses. The authors show two case studies of different wind directions, and use coincident airborne remote sensing observations of plume extents to derive mixing ratios from the MAX-DOAS measurements.

The paper is concisely written, well-organized and logical. The figures are very clear

and easy to follow. Overall I found the paper interesting and recommend it be eventually published. I did find it somewhat lacking in a description of motivation for the work and its possible application. The method for deriving horizontal information from MAX-DOAS using onion peeling was previously demonstrated for an urban area, and this paper is now applying it to ship emissions. This seems useful in theory, but it's not clear how the information would be used. Without plume extent information, the VMRs are derived over a long path in Section 4. Section 5 uses the airborne information to derive more precise VMR inside the plume, but these airborne measurements are rare and not regular. What would be the purpose of the MAX-DOAS measurements over a long time period? Would they be useful for trends, emissions estimates, monitoring etc? How can this be accomplished without plume width information, and are there other sources of this information? Can better modeling of ship plumes and NOx chemistry improve the estimates?

Also, aerosols, plume height and a few other sources of errors are quickly mentioned in Section 3.1, and clouds are quickly mentioned in Section 4.1. However, there is no thorough quantitative error assessment. I think the error sources need to be discussed and quantified in more detail. If you don't want to get into clouds, at least mention that for now you will only consider and draw conclusions about clear days. Also, error sources for the the AirMAP measurements should be described. There is an uncertainty given, but it is not clear from where it is derived. There are many possible error sources (fitting uncertainty, surface albedo, profile shape, aerosols etc).

Specific points:

Page 2, Line 5: specify whether these are ship or land based in situ measurements

Page 3, Section 2.1: Mention temporaral resolution of measurements here

Page 4, Line 7: Not sure column amount is a concentration?

Page 8, Line 9: Not sure what you mean by "instrument measures in wind direction"

Page 8, Line 14: NO2 only increases up to a point...

Page 11, Figure 6/7: In situ value colour saturates. Please mention what is the value in the text if not planning to change the colour scale. Maybe you could include it in Figure 5 as a function of time?

Figures 6/7/8/9: I find the forward trajectory of the plumes a bit hard to interpret. What is the timescale on these? Do the black to grey values denote anything?

Figure 10 and discussion in Section 5.3: I find the discussion of plume height a bit confusing and how it is used in the airborne observations. The MAX-DOAS on the tower seems to measure above the ship according to Figure 10, and the plume is not at the surface in the figure. Is there an assumed start height of the plume above the ocean? The AirMAP instrument is measuring the column to the surface. Why is 500 m used for the AMF calculation and not 335 m? Do the 335 m and 500 m height box profiles include a constant VMR to the surface? I don't think different assumptions will change the results by much, but the description of profiles and relation to the figure could do with some clarity.

Figures 11 and 14: Why show VMR and not DSCD for the MAX-DOAS here? Even though the DSCD is very diluted over a large area, it would at least put the measurements in the same units for easier visual comparison.

Technical corrections:

Page 2, Line 14: change colon to semicolon

Page 3, Line 25 and 26: Change "in a" to "at a" in both cases

Page 13, Line 2: "lightboth" not a word

figure 10: change "not up to scale" to "not to scale"

Page 18, Line 3: change colon to semicolon

---

## Author Comment (AC1) · 24 Jul 2019

GENERAL COMMENTS

The manuscript "Studies of the horizontal inhomogeneities in NO2 concentrations above a shipping lane using ground-based MAX-DOAS and airborne imaging DOAS measurements" presents nicely and picturesque the onion peeling approach applied to measurements in the German bight, demonstrated on individual measurements. In the second part of the manuscript, the authors compare 2 specific measurement instances to airborne imaging DOAS measurements taken during the NOSE campaign. These two instances show well the validity of the onion peeling approach, qualitatively and quantitatively.

While this second part, the comparison with imaging doas, makes also quantitative estimates, the first part, showing two times example measurements within approximately 12 minutes in 5 different azimuth directions, stays very qualitative. It neither includes an estimation of errors by e.g. the negligence of the correction factor in the O4 scaling approach for the effective light path estimation, nor does it include an attempt of making use of some ancillary information about the plume using e.g. the STEM model (Jalkanen et al. 2009 acp 9209-9223 , 2012 acp 2641-2659) and some diffusion model to estimate plume width/ height.

The authors argue that the presented method is suited to measure concentrations when wind conditions are unsuitable for surface measurements which do not measure any enhanced concentration if the wind blows the plume away from the measurement station. However, they themselves mention that the measured concentration with the onion peeling approach is not representative of the in-plume concentration. The authors lack to investigate possibilities how to extract useful information using other available information (using more info from the AIS data in combination with the STEM model and better modelling of the plumes). No attempt was made to connect the measured concentration to the in-plume concentration in the first part of the paper. Even if this is not carried out, I strongly recommend the authors to think about ways how this could be done and at least describe what could be done. Without it, this method seems rather incomplete and its usefulness quite limited.

However, the paper is very well presented and shows that if complementary measurements are available, useful estimates about plume concentrations of individual plumes can be calculated and hence should be published! It remains nevertheless unclear what the main purpose of the measurements is if no connection can be made to the in-plume concentration. This could certainly be clarified better both in the introduction and in the conclusions. I recommend to either include attempts to extract more information about the actual plume concentrations by using more ancillary information in

the first part of the paper (basically describing Fig. 6, 7, 8 and 9) or skipping that part and only concentrating on the second part. In the current state, it is a long paper that, over big parts, is rather qualitative and does not give much quantitative information about plume concentrations and hence is not quite suited for measurements of ship emissions on its own.

First, we would like to thank Anonymous Referee #1 for his/her helpful comments, particularly concerning the suggestion for the inclusion of ship plume modeling.

We updated and enhanced the described method by incorporating ship plume modeling using a simple Gaussian plume model and combining it with the plume forward trajectories. The information about plume width and height retrieved from the model is then used to derive in-plume volume mixing ratios of $NO_2$ from the MAX-DOAS measurements without the need for the airborne imaging DOAS measurements. In the new version, the AirMAP measurements are now only used for validation. As a consequence, the structure and aim of the paper was adapted. Section 5 was completely rewritten (now: Section 4.4) and contains two parts: The first part contains a technical demonstration of the method to derive in-plume $NO_2$ VMRs from MAX-DOAS measurements for ships passing the instrument in a distance of several km. We decided to demonstrate the method on the measurements during the NOSE campaign shown in Fig. 10 (new: Fig. 8), as AirMAP measurements for validation are available for this day.

In the second part of Section 4.4, the AirMAP measurements are used for validating both the plume modeling and the MAX-DOAS results. The modeled plume location and shape (including the plume width) is compared to the AirMAP measurements. The vertical plume extent from the model is compared to the estimation from the MAX-DOAS vertical scan, which was already included in the previous version. As before, the approximate plume position retrieved with the onion peeling MAX-DOAS approach is compared to the AirMAP measurements. The in-plume $NO_2$ VMR derived from the MAX-DOAS measurements is now compared to the in-plume VMR computed for the AirMAP measurements with help of the modeled plume height.

We kept the general structure, as we think the order of the results facilitates comprehension by enabling the readers to go step-by-step from the more basic time-series plots to the complex map figures which contain a lot of information. Starting with the time-series showing the relation between DSCDs and path-averaged VMRs, then taking the step from the time-series to the map figures with colored lines representing the VMRs and path lengths (for northerly and southerly wind directions) and finally the step to the figures additionally including the AirMAP measurements showing two completely different quantities: for AirMAP vertical columns of $NO_2$, for MAX-DOAS path-averaged $NO_2$ VMRs.

We think that the inclusion of plume modeling allowing derivation of in-plume $NO_2$ VMRs from MAX-DOAS measurements without the need of airborne measurements makes the paper scientifically more relevant and the described method much more quantitative and the main purpose of the measurements becomes clearer.

Below, we reply point-by-point to the specific comments. As far as possible, we have considered the suggestions in the revised manuscript.

SPECIFIC COMMENTS

(a) The authors only mention the restrictions on sulphur emission by the MARPOL convention. However, this manuscript is about NO2, so it should probably also give the info on this. Something like... EU adaption of this in form of directive 2012/33/EU NOx emissions depends on the rated rotational speed of the engine crankshaft, implementation in 3 tiers, last one not yet implemented, shifted to ~2021

Thank you very much, we must have overlooked this. Of course, the manuscript should mention the MARPOL $NO_x$ emission regulations in the North and Baltic sea (N)ECA. A corresponding paragraph has been added to the text.

(b) In the last paragraph of Sect. 3.2, the authors mention the importance of NO to NO2 conversion. Maybe the authors can give some estimates on time- (and spatial) scales for the increase (probably depending on some "standard"(?) background O3 concentration).

Some unpublished measurements performed at another site under roughly similar conditions indicate that already after a few minutes, the fraction of $NO_2$ in the overall $NO_x$ is quite high. After 1 minute, the NO content on the overall $NO_x$ is below 60% for most ships (but up to 96 % for some), after 3 minutes it drops to values below 40% for most ships (but up to 70-80% for some). After 5 minutes it is below 25% for most ships and after 8-10 minutes it is below 20-30% for all ships. Of course, this depends on the ambient ozone concentration.

Middleton et al. (2007)[1] modeled the NO to $NO_2$ conversion in plumes at short ranges, depending on the $O_3$ concentration:

[Figure]

**[NO2]:[NO*x*] from Janssen Method: Ozone [O3] 20-150 ppb and wind speed u = 10 m/s**

u=10 m/s, [O3]=20 ppb    u=10 m/s, [O3]=35 ppb
u=10 m/s, [O3]=50 ppb    u=10 m/s, [O3]=100 ppb
u=10 m/s, [O3]=150 ppb

**Figure 1**
Plot of results for the yield or ratio [$NO_2$]:[$NO_x$] using the Janssen (Janssen 1986, Janssen *et al*. 1988) method of near-source diffusion-limited $O_3$, which soon becomes asymptotic to the photostationary state further from the source, after approximately 200 seconds of travel time (2 km downwind here). The effect of changes in $O_3$ concentration is shown. The wind speed alters the choice of curve according to travel time (distance/speed). (In these curves distance is plotted on the x-axis; empirical curves later in this report show $NO_x$ on the x-axis.)

The figure shows that both the steady state value of the $NO_2$ to $NO_x$ ratio as well the time until the steady state is reached depend on the $O_3$ concentration.

At our Neuwerk station, typical background $O_3$ volume mixing ratios in summer are in the range of 30 to 40 ppb, but can go up to 60-70 ppb or down to 20 ppb as well. Taking a closer look at the curves for 35 ppb and 50 ppb ambient $O_3$ in the figure, it can be seen that the steady state is predicted to be reached already after 3 to 4 minutes and in the steady state the fraction of $NO_2$ on the overall $NO_x$ is 65-70%. This fits quite well to our measurements mentioned above.

Meier (2018)[2] shows AirMAP $NO_2$ measurements during an overflight over a ship and its plume from the NOSE campaign on 21 August 2013. The across−plume integrated $NO_2$ VCD increases with flown distance from the ship overpass, stabilizing on a plateau at a distance of around 3 km. This 3 km are not the distance since emission, as the plume is moved by the wind during the time from ship overpass to this point. Taking the combination of plume forward trajectories and simple Gaussian plume model, the plume age at this point is estimated to be ~400 seconds or ~6.5 minutes, in which the emitted air parcels traveled a distance of ~1.5 km. This is in the same order of magnitude than the measurements and model results discussed above.

[1] Middleton, D. R., Luhana, L. and Sokhi, R. S.: Review of methods for NO to NO2 conversion in plumes at short ranges, Environment Agency, Bristol., 2007.

[2] Meier, A. C.: Measurements of Horizontal Trace Gas Distributions Using Airborne Imaging Differential Optical Absorption Spectroscopy, phd thesis, University of Bremen, Bremen., 2018.

To conclude, after a few minutes, at the latest after 10 minutes, the NO to $NO_2$ conversion reaches its steady state, depending on the ambient ozone concentration. As the plumes considered in the manuscript are usually older than 10 minutes, NO to $NO_2$ titration should not have a strong influence on the presented results.

A summary of this information was included into the last paragraph of Section 3.2.

Also, the authors mention in Sect. 3.3 that plume broadening and dilution over time is neglected. Maybe an order of magnitude estimation should be included and it should be outlined how this information can be used to extract more useful information from the measurements. Which of the two effects dominates for which time-scales? Maybe a reference to some dispersion models that include chemistry?

As outlined above, we have added plume modelling to the manuscript which addresses this point at least partly.

(c) In Sect. 3.3 it is stated that the initialization period is 90 minutes before the measurements. The big plume (roughly N-S direction) present in all panels in Figure 8 (and 9), originates, according to the authors (Sect. 4.3 3rd paragraph) from two coal-fired power plants in Wilhelmshaven, 50 km away. The authors estimate the plume age to be 110 minutes. However, this suggests that the initialization period needed to be larger than 90 minutes, otherwise the plume would not have had the time to travel that far. I think this should be clarified.

After double-checking the numbers, it turned out that 90 minutes is in fact not correct. The applied initialization time for both case studies is 180 minutes (3 hours). This was corrected in the manuscript.

(d) Regarding the plume trajectories, maybe the "apparent wind" approach as illustrated e.g. in Berg et al (2012, amt 1085-1098) should be referenced.

Done.

(e) It is mentioned that there are two stations measuring wind conditions, one on Neuwerk, one on Scharhörn. However, it is not clear whether the wind used for the calculation of the plume trajectories in e.g. Fig. 6 or Fig.8 is a simple average of the two, or if it depends on the position of the plume at every given moment which wind (some sort of spatial interpolation) is applied, or if only one is used. Please clarify.

Depending on data availability and data quality, either wind data from the Scharhörn or Neuwerk weather station was used. Added a hint to the source of the weather data at the respective places in the text.

(f) Regarding the author's comment to Fig. 6 panel 1 why the plume of the small ship is not seen: Maybe the authors could do a quick calculation which heights are seen at the expected distance of the plume (about from about 20–40 m ?). What do the authors find more likely? Maybe using the STEAM model for that particular ship, together with estimates for the dilution due to diffusion and NO to NO2 conversion the authors could approximate in plume concentration and exclude or not exclude their first alternative.

The plume model gives the following results for this situation:

- plume age: 700-800 seconds,
- plume width: 1200-1300m,
- plume height: ~400m (reaching down to the ground) for a stack height of ~25 meters

It is therefore not likely, that the MAX-DOAS did not measure through the plume. NO to $NO_2$ titration (see above) also tells us that $NO_x$ is (very) roughly 80% $NO_2$ at this plume age. So the second alternative is also not likely. It is more likely, that the plume from this relatively "small" ship, which is quite strongly dispersed already, is also strongly diluted. If the amount of $NO_x$ emitted by this ship is relatively small, this might not be visible in the MAX-DOAS measurements.

To take this into account, the sentence was rewritten and now reads: "The fact that the plume from the smaller ship shows up only slightly in the measurements might be due to low emissions from this comparatively small ship and the dilution of the already strongly dispersed plume, as the plume model predicts a vertical extent of the plume of ~400 m and a plume width of 1200-1300 m at a plume age of 700-800 seconds."

(g) Can the authors comment on the effect on the MAXDOAS results when the plume is over the instrument, as also indicated by high in-situ measurements? Does this lead to cancelling effects or is the vertical extend of the plume negligible comparable to the horizontal?

This can be investigated by looking at the zenith sky measurements. For this, a different type of DOAS fit had to be done, using a noon reference[3] instead of the sequential reference[4] spectrum used in this study. The results can be seen in the following figure, showing the UV $NO_2$ DSCD for both off-axis measurements (0.5°, 1.5°, 2.5°, 3.5°, 4.5° and 30° elevation) in the 335° azimuth direction and zenith sky measurements (90° elevation) for the same day as in Fig. 6 (new: Fig. 4).

[Figure]

The zenith sky measurements (orange line, close to zero) indeed show enhanced values on this day, around 10:20 UTC and also around 12:50 UTC, at the very time of the respective situation in Fig. 6 (new: Fig. 4) when the plumes are reaching the radar tower/are over the instrument and in-situ values are high. At 12:50-12:53 UTC, a maximum $NO_2$ DSCD of $4 \times 10^{15}$ molecules/cm$^2$ is measured. Nevertheless, compared to the measurements in the 0.5° elevation used for the onion peeling reaching up to $1.3 \times 10^{17}$ molecules/cm$^2$ and in 1.5° elevation reaching even up to $1.7 \times 10^{17}$ molecules/cm$^2$, this number is small. While the vertical extent of the plume is certainly not negligible (the model says 400-450 m), it seems to have only a small influence on the zenith sky measurements because of the short vertical light path. This $NO_2$ enhancement in the zenith sky measurements definitely causes a canceling effect when using the sequential reference, but the overall impact seems to be negligible small (2 to 4 %). We added the following sentences to the text:

"A small $NO_2$ enhancement of $4 \times 10^{15}$ molec/cm$^{-2}$ is seen in the zenith sky measurements around 12:50 UTC, which is gone at 12:55 UTC, indicating that at least part of the plume was located above the MAX-DOAS instrument. As the zenith sky measurements are used as a sequential reference for the off-axis measurements, this causes a small canceling effect when using the sequential reference. As off-axis DSCDs are on the order of $1 \times 10^{17}$ molec/cm$^{-2}$ reaching up to $1.4 \times 10^{17}$ molec/cm$^{-2}$ as can be seen from Fig. 3, the overall impact on the path averaged VMRs is very small, on the order of 2 to 4 %."
* * *
[3] one zenith spectrum at noon is taken as the reference spectrum for this day

[4] for each spectrum a close-in-time interpolated zenith spectrum is taken as a reference

(h) Fig6, panel 10: The plume from the big ship cannot have yet reached the VIS-only region (Delta L). However, compared to the measurement 4 minutes before, the VMR seems to have increased by around 1.5 ppb. Any suggestion why?

There is definitely an increase in the $NO_2$ DSCDs measured in the visible from 0.9 ppb to 1.6 ppb. The following figure shows both the off-axis measurements (0.5° and 2.5° elevation) for this viewing direction (310° azimuth) and the zenith sky measurements in the visible:

[Figure]

Possible reasons could be emissions from another ship, the AIS signal of which was not received or emissions from a land-based source somewhere at the west-coast of Schleswig-Holstein, which is quite unlikely. More likely, however, is an uncertainty (overestimation) in the path length estimation due to negligence of the correction factor (see next question below), meaning that this increase is already coming from the ship crossing the line-of-sight in Panel 15. As can be seen from the measurements in the figure shown above, the $NO_2$ values drop quickly to ambient background values 4 minutes after Panel 15 when ship and plume have fully crossed the LOS and are now westward of it.

(i) Similarly, panel 15 seem to indicate a larger intersect of the plume with the viewingdirection for the UV region than for the VIS-only region. Still it looks like (as mentioned below, maybe not the best choice of colour map) the VIS-only region has a much higher average VMR. Probably this is due to the effect of overestimated length (due to negligence of correction factor as mentioned by the authors) and hence more of the intersection is in the UV-only path?

In fact this might be an example showing the uncertainty (overestimation) in the path length estimation due to negligence of the correction factor. The following paragraph was added to the manuscript:

"In Panel 15 the larger ship has moved further away from the instrument, leading for the first time in this sequence to a higher concentration on $\Delta L$, far away from the instrument, than close by. Comparing the locations of the MAX-DOAS paths with the ship position and modeled plume in detail, however, indicates a much larger intersect of the plume with the UV path than with $\Delta L$. This might be an example showing the uncertainty (overestimation) in the path length estimation due to negligence of the correction factor as discussed in Section 3.1."

(j) The first two (not numbered) equations seem to indicate that the air density in fact cancels out in the authors approach to estimate the VMR since only surface values for concentrations?

The air density does not cancel out, because the number density of $O_4$ contains the square of the number density of air, as $n_{O4} = (0.21 \cdot n_{O2})^2$. The first equation is therefore:

$$L = \frac{SCD_{O_4,horiz} - SCD_{O_4,zenith}}{n_{O_4}} = \frac{DSCD_{O_4}}{n_{O_4}} = \frac{DSCD_{O_4}}{(n_{O_2})^2} = \frac{DSCD_{O_4}}{(0.20942 \cdot n_{air})^2}$$

Inserting the first equation for the path length into the second equation yields:

$$
\begin{aligned}
\mathrm{VMR_{NO_2}} &= \frac{\mathrm{SCD_{NO_2,horiz}} - \mathrm{SCD_{NO_2,zenith}}}{L \cdot n_\mathrm{air}} = \frac{\mathrm{DSCD_{NO_2}}}{L \cdot n_\mathrm{air}} \\
&= \frac{\mathrm{DSCD_{NO_2}} \cdot (0.20942 \cdot n_\mathrm{air})^2}{\mathrm{DSCD_{O_4}} \cdot n_\mathrm{air}} = \frac{\mathrm{DSCD_{NO_2}} \cdot 0.20942^2 \cdot n_\mathrm{air}}{\mathrm{DSCD_{O_4}}}
\end{aligned}
$$

Note that the units, at a first glance maybe counter-intuitive, do fit together here: $[\mathrm{DSCD_{NO_2}}]$ = molecules/cm$^2$, $[n_\mathrm{air}]$ = molecules/cm$^3$, $[\mathrm{DSCD_{O_4}}]$ = molecules$^2$/cm$^5$,

fitting to the unit-less quantity VMR.

(k) The title suggests a more "equal weight" between the two methods in terms of "being presented". However, the imaging approach seems to be merely used for validation and is not presented as such, since this is done in a different publication. Maybe the title should reflect this.

The title was changed to reflect this.

(l) The authors conclude in their last sentence of the manuscript that this approach canbe successfully applied to ship emission measurements. Nowhere in the paper is an estimation of the ship emission presented. I advice the authors to delete or reformulate this sentence.

The sentence was reformulated and now reads: "To conclude, the presented measurements provide a real world demonstration that the onion peeling approach works for MAX-DOAS measurements and can successfully be applied to investigate air pollution by ships and to derive in-plume NO$_2$ volume mixing ratios for ships passing the instrument in a distance of several km."

TECHNICAL CORRECTIONS AND SUGGESTIONS

(a) page 2, line 31: ... is of (not on) the order of....

Done.

(b) page 5/6: 2 equations on these sides are not labelled. I think all equations shouldbe labelled.

Corrected.

(c) page 8, last sentence of penultimate paragraph: This is a really confusing sentence. I would probably reformulate to something like: "The movement of the ship together with the measured wind results in an apparent wind direction very different from the measured wind direction. Therefore, a measurement along the measured wind direction does not in genereal correspond to measurements along the plume".

Reformulated the sentence to: "But as the movement of the ship together with the measured wind can result in an apparent wind direction very different from the measured wind direction a measurement along the measured wind direction does not in general correspond to measurements along the plume. "

(d) Fig. 3,4 and 5: For easier reference, it might be a good idea to bundle those intoone figure with 3 vertically aligned panels.

Done. The three figures are now bundled into one figure with three panels.

(e) Fig 6 and 8: maybe a length scale would be nice to include.

Good idea. A length scale was included into the figures. To not overly clutter the small maps, it is shown only in the first panel of each figure.

Also, I suggest to label the viewing directions on the right-hand side on each row.

A very good idea! As suggested, I included labels for the azimuthal viewing directions on the RHS.

I am not sure if a jet-like colour scale is the best choice. The gnuplot type one used in Fig. 1 or viridis or any other colour scale that is monotone in lightness (The first and the last comment also hold for Fig. 7 and 9) would be better. However, maybe that is just something the authors can keep in mind for the next publication.

As you have already noticed, "Plasma", one of the new perceptually uniform sequential colormaps introduced with *Python* package *Matplotlib* version 2, was used for the ship traffic density map in Fig. 1. For the onion-peeling maps (e.g. former Fig. 6-9, new: Fig. 4-7), the usage of the colormap "Viridis" was tried before but turned out to be problematic, because in the printed version of the figures, the dynamic range of the colormap was too small. Here as an example the scanned version of a print out of Figure 9 (new: Fig. 7):

[Figure]

In the printed version, $NO_2$ VMRs between ~0.6 and ~1.4 ppb have virtually the same color shade and are undistinguishable. An adjustment of the colorscale in this figure only using a smaller range of values makes the colorscale inconsistent with Fig. 8 (new: Fig. 6), where nearly the full range is needed. So we decided to use the colormap "jet", even though we are aware of the disadvantages of jet-like colormaps and the accompanying problems and try to avoid it as much as possible. However, in the last figures where MAX-DOAS and AirMAP measurements are shown in the same plot, we changed the AirMAP color-scale to viridis to better distinguish visually between the two.

(f) page 13, line 2: "lightboth"??

Corrected.

(g) page 14, Sect. 4.3, second paragraph: Figure 8... (not Figure 6)

Corrected.

(h) Figures 1,6,7,8, and 9: Can the authors quickly state why Nigelhörn and Scharhörn got merged into one island in their map?

This is an interesting point. The coastlines included in the maps are shapefiles produced from OpenStreetMap (OSM) coastline data. The maps were created with the python package "Basemap" which incorporates another coastline data set, the GSHHG (Global Self-consistent, Hierarchical, High-resolution Geography Database), formerly known as GSHHS (Global Self-consistent, Hierarchical, High-resolution Shorelines). Coastlines can also be extracted from the GADM Database of Global Administrative Areas. The following three maps show the outlines of the islands Neuwerk, Scharhörn and Nigehörn in the different data sets:

[Figure]

In the OpenStreetMap data, the two islands Nigehörn and Scharhörn are connected (I also wondered about that). In the GSHHG data set (this is already the highest resolution), they do look very strange. In the GADM data set, they are separated, but the problem with using the GADM data set is, that the western coastline of the island Neuwerk has a very strange shape, which does not reflect the reality.

So, which data set is better? Some deeper research revealed that the answer to this question is not so clear. Off course, when the artificial island Nigehörn was created in 1989 by deposition of 1.2 million cubic metres of sand on the "Scharhörnplate" sandbank, both were separate islands. But in the following decades, Nigehörn naturally grew on the wadden sea side, towards the east, from 30 ha in 1989 to about 50 ha in 2004. Both islands are growing together (coalesce) and the wadden sea ground between the islands on the Scharhörnplate is growing in height. Since a few years the islands are somehow "connected" by a growing salt marsh (salt meadow), which is largely safe from flooding during high tide. In the future, the islands will grow together.

(Sources:
https://de.wikipedia.org/wiki/Nigeh%C3%B6rn,
https://www.nationalpark-wattenmeer.de/hh/luftbildpanoramen/spots/scharhoern-ost03,
http://nationale-naturlandschaften.de/gebiete/nationalpark-hamburgisches-wattenmeer/,
all visited on 18.05.2019)

This can even be seen on satellite maps:

[Figure]

So the question whether the islands Nigehörn and Scharhörn should be separate on a map or not, cannot be answered so clearly. The shape of Scharhörn and Nigehörn seen in the satellite map fits nicely to the OSM coastline data and the OSM coastline has the best representation of the Neuwerk coastline, too. This is why it was chosen for the map plots.

---

## Author Comment (AC2) · 24 Jul 2019

This paper presents measurements of NO2 from ship emissions in the German Bight using MAX-DOAS instrument, and shows that horizontal information on the NO2 distribution can be derived using an onion peeling method with NO2 slant columns derived separately in the UV and visible, which are observing slightly different air masses. The authors show two case studies of different wind directions, and use coincident airborne remote sensing observations of plume extents to derive mixing ratios from the MAXDOAS measurements.

The paper is concisely written, well-organized and logical. The figures are very clear and easy to follow. Overall I found the paper interesting and recommend it be eventually published. I did find it somewhat lacking in a description of motivation for the work and its possible application. The method for deriving horizontal information from MAXDOAS using onion peeling was previously demonstrated for an urban area, and this paper is now applying it to ship emissions. This seems useful in theory, but it's not clear how the information would be used. Without plume extent information, the VMRs are derived over a long path in Section 4. Section 5 uses the airborne information to derive more precise VMR inside the plume, but these airborne measurements are rare and not regular. What would be the purpose of the MAX-DOAS measurements over a long time period? Would they be useful for trends, emissions estimates, monitoring etc? How can this be accomplished without plume width information, and are there other sources of this information? Can better modeling of ship plumes and NOx chemistry improve the estimates?

Also, aerosols, plume height and a few other sources of errors are quickly mentioned in Section 3.1, and clouds are quickly mentioned in Section 4.1. However, there is no thorough quantitative error assessment. I think the error sources need to be discussed and quantified in more detail. If you don't want to get into clouds, at least mention that for now you will only consider and draw conclusions about clear days.

Also, error sources for the the AirMAP measurements should be described. There is an uncertainty given, but it is not clear from where it is derived. There are many possible error sources (fitting uncertainty, surface albedo, profile shape, aerosols etc).

First, we would like to thank Anonymous Referee #1 for his/her helpful comments, particularly concerning the suggestion for the inclusion of ship plume modeling.

We updated and enhanced the described method by incorporating ship plume modeling using a simple Gaussian plume model and combining it with the plume forward trajectories. The information about plume width and height retrieved from the model is then used to derive in-plume volume mixing ratios of NO$_2$ from the MAX-DOAS measurements without the need for the airborne imaging DOAS measurements. In the new version, the AirMAP measurements are now only used for validation. As a consequence, the structure and aim of the paper was adapted. Section 5 was completely rewritten (now: Section 4.4) and contains two parts: The first part contains a technical demonstration of the method to derive in-plume NO$_2$ VMRs from MAX-DOAS measurements for ships passing the instrument in a distance of several km. We decided to demonstrate the method on the measurements during the NOSE campaign shown in Fig. 10 (new: Fig. 8), as AirMAP measurements for validation are available for this day.

In the second part of Section 4.4, the AirMAP measurements are used for validating both the plume modeling and the MAX-DOAS results. The modeled plume location and shape (including the plume width) is compared to the AirMAP measurements. The vertical plume extent from the model is compared to the estimation from the MAX-DOAS vertical scan, which was already included in the previous version. As before, the approximate plume position retrieved with the onion peeling MAX-DOAS approach is compared to the AirMAP measurements. The in-plume $NO_2$ VMR derived from the MAX-DOAS measurements is now compared to the in-plume VMR computed for the AirMAP measurements with help of the modeled plume height.

We kept the general structure, as we think the order of the results facilitates comprehension by enabling the readers to go step-by-step from the more basic time-series plots to the complex map figures which contain a lot of information. Starting with the time-series showing the relation between DSCDs and path-averaged VMRs, then taking the step from the time-series to the map figures with colored lines representing the VMRs and path lengths (for northerly and southerly wind directions) and finally the step to the figures additionally including the AirMAP measurements showing two completely different quantities: for AirMAP vertical columns of $NO_2$, for MAX-DOAS path-averaged $NO_2$ VMRs.

We think that the inclusion of plume modeling allowing derivation of in-plume $NO_2$ VMRs from MAX-DOAS measurements without the need of airborne measurements makes the paper scientifically more relevant and the described method much more quantitative and the main purpose of the measurements becomes clearer.

Adding $NO_x$ chemistry to the plume model would certainly improve the results but would also be more challenging. As the plumes measured in the study are mostly rather old (plume age usually > 10 minutes), we expect most of the NO to be already converted to $NO_2$. Some unpublished measurements performed at another site under roughly similar conditions indicate that already after a few minutes, the fraction of $NO_2$ in the overall $NO_x$ is quite high. After 1 minute, the NO content on the overall $NO_x$ is below 60% for most ships (but up to 96 % for some), after 3 minutes it drops to values below 40% for most ships (but up to 70-80% for some). After 5 minutes it is below 25% for most ships and after 8-10 minutes it is below 20-30% for all ships. Of course, this depends on the ambient ozone concentration.

Middleton et al. (2007)[1] modeled the NO to NO₂ conversion in plumes at short ranges, depending on the $O_3$ concentration:

[Figure]

**[NO2]:[NO*x*] from Janssen Method: Ozone [O3] 20-150 ppb and wind speed u = 10 m/s**

Figure from:
Middleton et al. (2007)

| | | |
|---|---|---|
| —■— u=10 m/s, [O3]=20 ppb | —— u=10 m/s, [O3]=35 ppb | |
| —▲— u=10 m/s, [O3]=50 ppb | —■— u=10 m/s, [O3]=100 ppb | |
| —✳— u=10 m/s, [O3]=150 ppb | | |

**Figure 1**
Plot of results for the yield or ratio [$NO_2$]:[$NO_x$] using the Janssen (Janssen 1986, Janssen *et al.* 1988) method of near-source diffusion-limited $O_3$, which soon becomes asymptotic to the photostationary state further from the source, after approximately 200 seconds of travel time (2 km downwind here). The effect of changes in $O_3$ concentration is shown. The wind speed alters the choice of curve according to travel time (distance/speed). (In these curves distance is plotted on the x-axis; empirical curves later in this report show $NO_x$ on the x-axis.)

The figure shows that both the steady state value of the $NO_2$ to $NO_x$ ratio as well the time until the steady state is reached depend on the $O_3$ concentration.

At our Neuwerk station, typical background $O_3$ volume mixing ratios in summer are in the range of 30 to 40 ppb, but can go up to 60-70 ppb or down to 20 ppb as well. Taking a closer look at the curves for 35 ppb and 50 ppb ambient $O_3$ in the figure, it can be seen that the steady state is predicted to be reached already after 3 to 4 minutes and in the steady state the fraction of $NO_2$ on the overall $NO_x$ is 65-70%. This fits quite well to our measurements mentioned above.

Meier (2018)[2] shows AirMAP $NO_2$ measurements during an overflight over a ship and its plume from the NOSE campaign on 21 August 2013. The across–plume integrated $NO_2$ VCD increases with flown distance from the ship overpass, stabilizing on a plateau at a distance of around 3 km. This 3 km are not the distance since emission, as the plume is moved by the wind during the time from ship overpass to this point. Taking the combination of plume forward trajectories and simple Gaussian plume model, the plume age at this point is estimated to be ~400 seconds or ~6.5 minutes, in which the emitted air parcels traveled a distance of ~1.5 km. This is in the same order of magnitude than the measurements and model results discussed above.

To conclude, after a few minutes, at the latest after 10 minutes, the NO to $NO_2$ conversion reaches its steady state, depending on the ambient ozone concentration.  As the plumes

[1] Middleton, D. R., Luhana, L. and Sokhi, R. S.: Review of methods for NO to NO2 conversion in plumes at short ranges, Environment Agency, Bristol., 2007.

[2] Meier, A. C.: Measurements of Horizontal Trace Gas Distributions Using Airborne Imaging Differential Optical Absorption Spectroscopy, phd thesis, University of Bremen, Bremen., 2018.

considered in the manuscript are usually older than 10 minutes, NO to $NO_2$ titration should not have a strong influence on the presented results. However, for a potential next step, the derivation of $NO_x$ emission factors, accurate modeling of the $NO$-to-$NO_2$-titration would be important.

Regarding clouds: The following sentence was added to the manuscript: "In the following, only clear sky days or measurements under cloud free conditions are considered." and the whole paragraph was moved to Section 3.1.

As requested, a few words on possible error sources for the AirMAP measurements were also added.

Below, we reply point-by-point to the specific comments. As far as possible, we have considered the suggestions in the revised manuscript.

Specific points:

Page 2, Line 5: specify whether these are ship or land based in situ measurements

Both ship-borne and land-based in-situ measurements of shipping emissions are common. There are even airborne measurements, e.g. by Beecken et al. (2014)[3] and Balzani Lööv et al. (2014)[4].

Changed the sentence from

"Most measurements of pollution are performed with in-situ instrumentation, and this includes monitoring of the effect of ship emissions."

to

"Most measurements of **air** pollution are performed with in-situ instrumentation, and this includes monitoring of the effect of ship emissions, **which is usually performed with either land-based or shipborne in situ measurements.**"

Page 3, Section 2.1: Mention temporaral resolution of measurements here

We added the following sentences to Section 2.1:
"The total exposure time (or integration time) per measurement is 10 seconds for off-axis measurements and 20 seconds for zenith sky reference measurements. A new azimuthal measurement in one of the five different directions (see Section 2.2 and  Fig. 1) starts about every 30 seconds. The measurement sequence is intermitted by a vertical scan in the main direction (335° azimuth) and a zenith sky measurement, both together taking in total around 90 seconds. The temporal resolution for one viewing direction, i.e. the time until the same azimuthal direction is probed again, is around 4 minutes. "

Page 4, Line 7: Not sure column amount is a concentration?

A column amount is not a concentration, but integrating a concentration along a certain light path delivers a column amount. Changed the structure of the sentence from "The quantity retrieved from DOAS measurements is the concentration of an absorber integrated along the atmospheric light path, the so-called slant column density (SCD)."

to

"The quantity retrieved from DOAS measurements is **the so-called slant column density (SCD), the integrated concentration of an absorber along the atmospheric light path.**"
to make it more precise.

3   Beecken, J., Mellqvist, J., Salo, K., Ekholm, J., and Jalkanen, J.-P.: Airborne emission measurements of SO2 , NOx and particles from individual ships using a sniffer technique, Atmos. Meas. Tech., 7, 1957-1968, https://doi.org/10.5194/amt-7-1957-2014, 2014.

4   Balzani Lööv, J. M., Alfoldy, B., Gast, L. F. L., Hjorth, J., Lagler, F., Mellqvist, J., Beecken, J., Berg, N., Duyzer, J., Westrate, H., Swart, D. P. J., Berkhout, A. J. C., Jalkanen, J.-P., Prata, A. J., van der Hoff, G. R., and Borowiak, A.: Field test of available methods to measure remotely SOx and NOx emissions from ships, Atmos. Meas. Tech., 7, 2597-2613, https://doi.org/10.5194/amt-7-2597-2014, 2014.

Page 8, Line 9: Not sure what you mean by "instrument measures in wind direction"

The whole sentence was reformulated to: "But as the movement of the ship together with the measured wind can result in an apparent wind direction very different from the measured wind direction, a measurement along the measured wind direction (windward, i.e. pointing anti-parallel to the wind vector) does not in general correspond to a measurement along the plume. "

Page 8, Line 14: NO2 only increases up to a point…

Yes, of course, thanks for noticing. Corrected to:

"Therefore, the $NO_2$ signal increases **(up to a point)** with distance from the ship and, depending on the wind direction, with distance from the ship track."

Page 11, Figure 6/7: In situ value colour saturates. Please mention what is the value in the text if not planning to change the colour scale. Maybe you could include it in Figure 5 as a function of time?

Good suggestion, as the information that the in situ instrument in fact measured two overlapping plumes got lost in the saturated color scale before. We included the measured in-situ values in the text and additionally pointed out that the in situ instrument measured two overlapping plumes.

Adding the in-situ curve to Figure 5, however, does not work, as the in-situ instrument measures much higher values, in this case for example reaching nearly 10 ppb and extending the y-axis range to account for the much higher in-situ values would substantially decrease the dynamics of the MAX-DOAS curves which is needed here for distinguishing the different cases.

Figures 6/7/8/9: I find the forward trajectory of the plumes a bit hard to interpret. What is the timescale on these? Do the black to grey values denote anything?

The lightness of the gray shading denotes the age of the plume. A colorbar showing the relationship between the lightness of the gray shading and the plume age was included in all plots showing modelled plumes.

Figure 10 and discussion in Section 5.3: I find the discussion of plume height a bit confusing and how it is used in the airborne observations. The MAX-DOAS on the tower seems to measure above the ship according to Figure 10, and the plume is not at the surface in the figure. Is there an assumed start height of the plume above the ocean?

As the MAX-DOAS measures down to the sea surface, the plume is assumed to reach down to the surface as well. The plume modelling supports this assumption. We added a hint on this to the text.

We also added the information that the heights are not to scale to the caption of Figure 10.

The AirMAP instrument is measuring the column to the surface. Why is 500 m used for the AMF calculation and not 335 m? Do the 335 m and 500 m height box profiles include a constant VMR to the surface? I don't think different assumptions will change the results by much, but the description of profiles and relation to the figure could do with some clarity.

A 335m box profile could have been used, but only for this specific ship. The vertical extent of the other plumes in the figure are certainly different from this, probably larger, as the plumes are older. 500m seemed like a good first guess for all the ships. The correction to the measured/modeled plume height (here 335m or 320m, respectively) is done in the computation of the in plume VMR. And yes, the box profile assumes constant $NO_2$ up to 500m. We added this information to the manuscript and the respective paragraph now reads:

"For the retrieval of $NO_2$ vertical column densities, air mass factors were calculated for an $NO_2$ box profile **assuming constant NO₂** in the lowest 500 m, in an atmosphere without aerosols and for a constant surface reflectance of 0.05. **This box profile height is an educated guess on an upper limit for the typical vertical plume extent for older ship plumes, which the plume modeling has proven to be in the right order of magnitude.**"

Figures 11 and 14: Why show VMR and not DSCD for the MAX-DOAS here? Even though the DSCD is very diluted over a large area, it would at least put the measurements in the same units for easier visual comparison.

Granted, but the DSCD along the path difference, $\Delta DSCD = DSCD_{vis} - DSCD_{UV}$, is smaller than the DSCD along the UV path, $DSCD_{UV}$, and this reverses the situation in the figure: The higher $NO_2$ value is no longer shown along $\Delta L$, where the plume is located, but close to the instrument along $L_{UV}$, and much lower $NO_2$ is shown along $\Delta L$ where the plume is. This is rather unintuitive, as shown below, so we would prefer to keep the plotted quantities as is.

Existing figure showing MAX-DOAS **VMR**:          Figure showing MAX-DOAS **DSCD** instead:

[Figure]

To better visually discriminate the two measurements, we changed the color-scale of the AirMAP measurements to *viridis*, one of the new perceptually uniform sequential colormaps introduced with *Python* package *Matplotlib* version 2.

Technical corrections:

Page 2, Line 14: change colon to semicolon

Done.

Page 3, Line 25 and 26: Change "in a" to "at a" in both cases

Done.

Page 13, Line 2: "lightboth" not a word

Changed "concentrations on all lightboth path segments" to "concentrations on both path segments"

figure 10: change "not up to scale" to "not to scale"

Done.

Page 18, Line 3: change colon to semicolon

Done.